# Tolerance for democratic norm violations increases when sincerity replaces accuracy as a marker of honesty
Kiia Jasmin Alexandra Huttunen [1] ✉ & Stephan Lewandowsky [1,2]

People's subjective conceptions of truth and honesty have undergone significant changes in recent decades. Parts of society increasingly favour the sincere expression of personal belief, however inaccurate, as a marker of honesty over verifiable facts. At the same time, political elites in many democracies have been increasingly violating democratic norms. Those violations have been identified as a major contributor to democratic backsliding, highlighting the need for a thorough examination of the nexus between democratic norm violations and conceptions of honesty. We present a series of four preregistered experiments (total $n = 1537$) that examined the conditions under which people acquiesce to democratic norm violations and politicians' dishonesty. We find that when participants are asked to take a perspective of honesty that emphasises sincerity over accuracy, which we call "belief-speaking", they are more willing to accept norm violations by politicians than if participants take a perspective that emphasizes accuracy as a criterion for honesty, which we call "fact-speaking". When a fictitious politician is presented as telling untruths, tolerance of norm violations is reduced compared to when the politician is presented as truthful. The findings highlight the need to develop a better understanding of how individuals interpret and respond to political leaders' behaviours, especially in a context of widespread democratic backsliding.

Democratic governance is a balancing act between respecting the will of the majority while also protecting minority views. No one is born as a good citizen and no state is born as a democracy—both people and society are in constant flux and democracy cannot be taken for granted. Threats against democracy are manifold, and many countries are currently experiencing democratic backsliding[1–3]. The global rise of autocracy and the extreme right under the banner of "populism" is well-documented[4,5] and poses significant challenges to democratic governance and societal stability.

A defining attribute of right-wing populism is that it considers society to be divided into two implacably opposed camps, namely "the people", who are virtuous and whose will must prevail, and "the elites", who are corrupt and seek to obstruct the will of the people[6]. By rejecting the idea of a plurality of legitimate opinions and, in most cases, concentrating power in the hands of a single leader, right-wing populism erodes the foundations of democracy[6,7]. Some researchers [e.g., ref. 8] argue that not all forms of populism are right-wing, and that populism can also have beneficial consequences for society, for example through increased political participation. Similarly, people on both sides of the left-right political spectrum have been shown to engage in motivated reasoning congruent with their prior beliefs[9,10]. Research into the differences and similarities between left-wing

and right-wing manifestations of political cognition are valuable, but those results are not directly relevant to the present paper. Here we focus exclusively on right-wing extremism under the banner of populism, which has been identified as the greatest risk to democracy, at least in Europe[11,12].

Right-wing populists present themselves as the true champion of the people, the outsider-on-the-inside, who dares to take on the "elite". The mythical "elite" is redefined as needed for the populist leader to retain their claim to be on the side of the "people"—even when they hold power and are the elite, populist leaders will conjure up a shadowy "deep state", intellectuals, public-health officials, the judiciary, or any other presumed or actual opponent as the elite or "enemies of the people"[13].

This stance has two major consequences. First, the violation of institutional norms, such as those of mutual tolerance, institutional forbearance (i.e., not merely assessing the legality of an action but also its normative acceptability), or judicial independence, becomes a sign of distinction rather than a stigma[7,14–16]. For example, when British Prime Minister Boris Johnson suspended the U.K. Parliament for five weeks in 2019, thus eliminating Parliament's ability to scrutinise the government's Brexit policy, he was cheered on by much of the tabloid media as taking on the "elite" [e.g., ref. 17]. A body of research has confirmed that democratic norm violations

[1]University of Bristol, Bristol, UK. [2]University of Potsdam, Potsdam, Germany. ✉e-mail: Kiia.Huttunen@Bristol.ac.uk

and widespread dishonesty are nearly inescapable consequences of right-wing populism [e.g., refs. 18,19].

Second, right-wing populists often appeal to the "common sense" of "the people" as a source of truth, rejecting "elitist" notions such as evidence and expertise[20]. The right-wing populist leader dares to do what others do not[21]. For them, truth-telling based on facts and expertise constitutes a form of control and they instead consider truth as a political endeavour necessitating loyalty to leaders and causes[22].

The deliberate rejection of expertise and evidence by leaders is facilitated by the well-established fact that people with a conservative or right-wing political orientation tend to rely more on intuition than evidence to determine what they consider to be the truth[23]. Conservatives have a preference for heuristic cues and are more likely to rely on anecdotal evidence over detailed, fact-based, analysis of a given topic, in particular if the evidence is in opposition to their beliefs[24]. It has been argued that when this intuition-based view of the world is challenged, conservatives may end up supporting anti-democratic actions and candidates[11].

Setting aside political orientation, little is known about why the public would tolerate politicians' norm violations and widespread dishonesty. For example, while one might expect that politicians who are pervasive purveyors of misinformation suffer reputational costs, those costs are often not observed in experiments[25,26] or indeed at the ballot box[27,28]. Given the threats that norm violations by political elites pose to democratic stability, understanding public reactions to them is crucial[29]. To date, research has proceeded along two avenues to explain public acquiescence to dishonesty and norm violations.

One avenue has focused on the role of people's disgruntlement or disenfranchisement, two principal drivers of endorsement of populist parties[30–32]. To illustrate, in one experiment, lying politicians were found to gather support when at least one constituency saw the political system as flawed or felt excluded by it. By contrast, when the system was perceived as fair, lying politicians were not tolerated[33]. Politicians who lie pervasively may thus be perceived as strong leaders by their supporters; that is, as someone who can solve or win a perceived inter-group conflict[34,35]. Similarly, violation of the norm of civility has been found to be tolerated by people with populist attitudes[36]. In general, politicians who violate norms may be seen as more dominant and prestigious in situations of competition and are rewarded with more support from citizens[37–39]. It is unknown, however, which types of norm violations are more tolerated than others and whether there are specific preconditions that make norm violations by politicians more tolerable.

A second stream of research has focused on the role of different ontologies or subjective conceptions of truth and honesty[40–42]. Those different conceptions invariably cluster into one aspect involving intuition, feeling, or sincerity and another aspect involving evidence, facts, and accuracy. To illustrate, Aroyehun et al.[42] examined speeches in the U.S. Congress from 1879 until 2022 and discovered that the use of evidence-based language has steadily declined after reaching a peak in the mid-1970's. This shift from evidence to intuition in congressional rhetoric has been associated with an increase in partisan polarisation in Congress and society at large[42]. The shift from evidence to intuition is explainable within a right-wing populist logic, in which the honesty of communication is established more in an actor-based instead of a claim-based framework; there is a greater focus on the authenticity and sincerity of the person making a claim ("he says it how it is") than there is on the truthfulness of the claim itself. Within a democratic and pluralistic logic, by contrast, the speaker's sincerity is less relevant than their factual accuracy. Hence, as populism grows, so does reliance on intuition.

The two end points of a continuum from an emphasis on accuracy to reliance on sincerity as markers of honesty have been labelled, respectively, fact-speaking and belief-speaking[43–45]. Fact-speaking focuses on accuracy of claims and tends to make explicit reference to evidence. Belief-speaking, by contrast, focuses on the sincerity and authenticity of the speaker without references to external reality. An exclusive focus on sincerity of the speaker can contribute to the acceptance and further spread of misinformation[33,40,44,46]. This is buttressed by literature showing that conservatives—who are known to be more likely to prefer intuition over evidence—may display a backfire effect when misinformation is corrected. Rather than updating their views, they can become more likely to share misinformation than before the correction was presented[11]. However, political orientation and different subjective notions of truth are two distinct concepts that, despite appearing to be correlated, need to be disentangled and considered separately[47,48].

In this article, we focus on public acquiescence of norm violations and dishonesty by politicians from the perspective of the different notions of honesty just introduced. Unlike most previous research, our approach is not correlational but experimental: We administered an intervention in 4 experiments that directed people to take a fact-speaking or belief-speaking perspective on honesty. People were then presented with a vignette describing the actions of a hypothetical politician who was either norm abiding or norm violating, and who was either lying or truthful. Our primary measure was participants' tolerance for democratic norm violations. In addition, we also measured a variety of individual attributes as potential covariates, such as political orientation and epistemic preferences (see 'Methods').

Our over-arching expectation was that by orienting participants towards sincerity as the foundation of honesty, they would become more tolerant of politicians who are not telling the truth and who are violating democratic norms. Conversely, we expected participants who were oriented towards accuracy as the foundation of honesty to be less forgiving of lying and norm violations. We derived a number of hypotheses from this over-arching expectation during the course of this research, which were all pre-registered and are shown in Table 1. Particularly hypotheses H3 through H7 are all a slightly paraphrased version of each other, fine-tuned during the process of designing the studies and thus differing subtly. At their core, they are all looking at the same issue: the role of perspective-taking in participants' tolerance for democratic norm violations and in particular whether preference for sincerity is likely to make participants more tolerant of democratic norm violations, whereas preference for accuracy is likely to make them less tolerant.

## Methods

The hypotheses in Table 1 were tested in four preregistered studies that involved the same three fully-crossed experimental variables in a between-participants design. Figure 1 illustrates the common elements of the procedure for all experiments.

### Participants

The four studies received ethical approval from the University of Bristol School of Psychological Science Research Ethics Committee (project IDs 15550, 16111, 17717, and 23904). All participant recruitment was done through Prolific and participants were compensated at UK minimum wage or above. Study protocols and hypotheses were pre-registered on AsPredicted.

In all studies, participants who reported that they did not adopt the assigned perspective on honesty were excluded from the main analyses. This is known as a Per Protocol analysis which is recommended when adherence to experimental protocols is critical to infer the underlying psychological processes, as is the case here[49]. Parallel analyses involving all participants, known as an Intention to Treat analysis, are reported in Section 6 of the supplement. The Intention to Treat analyses yield qualitatively identical results to the main Per Protocol analyses.

For Experiment 1 (preregistered on 08/07/2023), we recruited a representative UK sample. In Experiment 2 (preregistered on 08/10/2023), we invited US based participants who had already participated in a previous study[50] that measured people's epistemic preferences. No additional ethics approval from a US institution was sought for this. This experiment was conducted to allow us to utilise the pre-existing Epistemic-Evidence Intuition Scale (E2IS; ref. 50) which measures people's epistemic preferences. For Experiment 3 (preregistered on 15/01/2024), a representative UK sample

**Table 1 | Hypotheses**

| ID | Hypothesis | Pre-registered in | Supported in | | | |
|----|-----------|-------------------|------|------|------|------|
| | | | Exp. 1 | Exp. 2 | Exp. 3 | Exp. 4 |
| H1 | Participants who prefer accuracy over sincerity are more likely to view accurate politicians more favourably than sincere politicians. | Exp. 1, 2 | ✗ | ✗ | ✗ | * |
| H2 | Participants who prefer sincerity over accuracy are more likely to view sincere politicians more favourably than accurate politicians. | Exp. 1, 2 | ✗ | ✗ | — | — |
| H3 | There is a difference in how the three norm-breaking behaviours—incitement, dishonesty, and expansion of power—are tolerated. | Exp. 1, 2, 3 | ✓ | ✓ | ✓ | — |
| H4 | Norm-breaking behaviour from sincere politicians is tolerated better than from accurate politicians. | Exp. 1 | ✓ | — | — | — |
| H5 | Participants who prefer accuracy over sincerity are likely to view an accurate politician less favorably if the politician is breaking democratic norms. | Exp. 3, 4 | — | — | ✓ | ✓ |
| H6 | Participants who prefer sincerity over accuracy are likely to view sincere politicians favorably even if the politician is breaking democratic norms. | Exp. 3, 4 | — | — | ✗ | ✗ |
| H7 | Preference for sincerity is likely to make participants more tolerant of democratic norm violations, whereas preference for accuracy is likely to make them less tolerant. | Exp. 4 | — | — | — | ✓ |
| H8 | Tolerance for democratic norm violations may be influenced by participants' political orientation. | Exp. 4 | — | — | — | ✓ |
| H9 | There will be a relationship between participants' political orientation and their epistemic preferences. | Exp. 4 | — | — | — | ✓ |

Note: ✓ = supported; ✗ = not supported; — = not tested; * = supported but not pre-registered for analyses.

was recruited. This experiment allowed us to combine our experimental intervention with measuring participants' pre-existing epistemic preferences and to test our pre-registered hypotheses on completely new data. Finally, at a reviewer's suggestion we conducted Experiment 4 (preregistered on 14/02/2025) to assess the potential role of political orientation and to buttress our conclusions. Prolific as a platform allows researchers to recruit participants based on their political orientation and for our total UK sample of 600 participants, we recruited 300 who had indicated to be on the left of the political spectrum and 300 on the right.

In all experiments, as per the preregistered protocol, participants who did not comply with the directed honesty perspective were removed. For all experiments, see Table 2 for preregistration IDs, links, sample sizes and a breakdown of demographics. No further demographic data from participants was collected. For a complete breakdown of participants per experimental condition, see Section 5 of the online supplement.

**Materials and procedure**

After reading the study description and providing their informed consent, all participants were presented with a vignette from Vargiu and Nai[46] describing the perspective of "Sam" (belief-speaking) or "Taylor" (fact-speaking); see Appendix A for the full vignettes. Participants were asked to adopt this perspective for the duration of the experiment and answer the questions from this perspective. Before proceeding with the experiment, the participant was asked to tick a box confirming that they had adopted to perspective—or had not, in which case their data was not included in the main analyses. In Experiment 1 only, the assignment was congruent with the participants' pre-existing perspective as measured with the "sincerity" and "accuracy" items from Vargiu and Nai[46]. The conceptualisation of "belief-speaking" and "fact-speaking" is rooted in previous work [e.g., ref. 45] and in the absence of relevant precedent, assigning participants to perspectives that aligned with their pre-existing, measured, perspective allowed us to maximise the effectiveness of the experimental manipulation. Once we confirmed that our instructions and situational manipulation were effective in Experiment 1, in the remaining studies assignment to perspective was random and did not consider participants' pre-existing epistemic preferences.

Following the perspective-taking step, participants were presented with a scenario developed for this study describing the actions of a hypothetical politician, Mr. Smith, who was always described as matching participants' assigned perspective (belief-speaking or fact-speaking). That is, Mr. Smith was described either to always speak his mind irrespective of the evidence available or to always stick to verifiable facts. In addition, the scenario instantiated two fully-crossed experimental factors, Normativity and Truthfulness. Mr Smith was presented to be either norm abiding or in violation of three democratic norms (by being dishonest, or by inciting violence, and pursuing an expansion of his power), and orthogonally, as telling the truth or lying (e.g., by using fabricated data to support proposed parliamentary changes). Participants were randomly assigned to a cell of the Normativity × Truthfulness design. Full scenarios are available at https://osf.io/tkr56/.

Participants could spend an unlimited amount of time reading the scenario. After presentation of the scenario, participants judged the acceptability of the three norm violations (see Section 4 of the online supplement for items) and then rated the perceived honesty and likeability of Mr. Smith by responding to questions about numerous different personal attributes; accuracy, authenticity, competence, genuineness, sincerity, honesty, truthfulness, prestige, considerateness, warmth, and likeability. In Experiment 1, an Exploratory Factor Analysis (EFA) was carried out to examine the associations among the 11 judged attributes. The items for prestige and considerateness were dropped from analyses for subsequent experiments because of low factor loadings (see "Data processing" for full description of the factor analyses.)

In Experiments 3 and 4, participants responded to the Epistemic Evidence Intuition Scale [E2IS; ref. 50], which measures people's epistemic preference for evidence or intuition. For Experiment 2, we invited

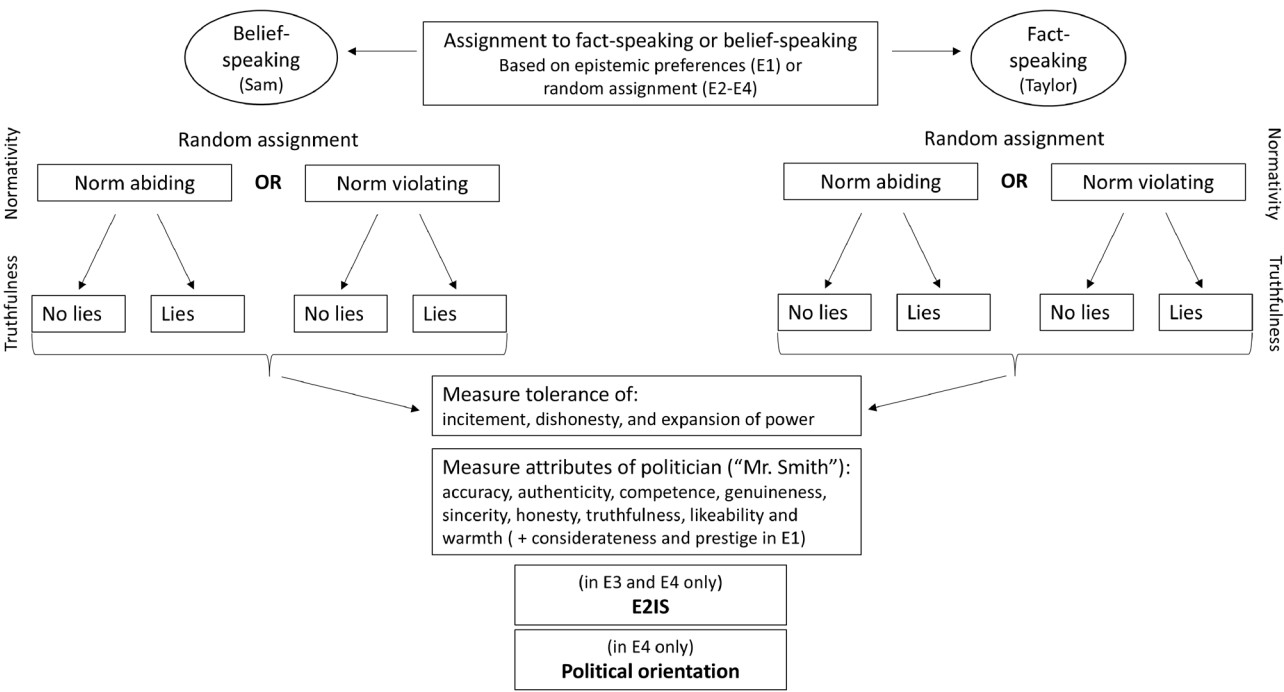

**Fig. 1 |** Visual representation of the experimental procedure in the four experiments.

## Table 2 | Overview of experiments

| | AsPredicted ID (link) | Country | Recruited N | Final n | $M_{age}(SD_{age})$ | Gender | | | |
|---|---|---|---|---|---|---|---|---|---|
| | | | | | | **Male** | **Female** | **Non-binary** | **Rather not say** |
| Experiment 1 | | | | | | | | | |
| | 137872 (aspredicted.org/4t3p5.pdf) | UK | 280 | 238 | 45.93 (15.80) | 114 | 122 | 0 | 2 |
| Experiment 2 | | | | | | | | | |
| | 146309 (aspredicted.org/xzhd-2hqk.pdf) | US | 365 | 294 | 41.90 (12.42) | 135 | 155 | 3 | 0 |
| Experiment 3 | | | | | | | | | |
| | 157986 (aspredicted.org/wx5g-xsph.pdf) | UK | 480 | 449 | 46.23 (15.69) | 218 | 230 | 1 | 0 |
| Experiment 4 | | | | | | | | | |
| | 212870 (aspredicted.org/4hqs-dk3w.pdf) | UK | 600 | 558 | 42.58 (14.45) | 282 | 311 | 5 | 2 |

Note: This table presents further information on all four experiments as well as demographic information.
Final n was acquired after removing participants who did not comply with the requested honesty perspective. Total n = 1537.

participants who had previously completed the E2IS in an unrelated study and merged their E2IS responses with the present data. The 21 items of the E2IS use a seven-point scale from "strongly disagree" to "strongly agree" and responses are combined into a single score that reflects a person's preference for evidence or intuition as a guide to ascertaining truth. The order of items was randomized with Qualtrics' built-in randomizer. Further information on the construction and validation of the E2IS is available in Section 7 of the online supplement.

In addition, in Experiment 4, political orientation was measured with an 11-point scale: "People sometimes use the labels 'left' or 'left-wing' and 'right' or 'right-wing' to describe political parties, party leaders, and political ideas. Using the 0 to 10 scale below, where the end marked 0 means left and the end marked 10 means right, where would you place yourself on this scale?"

### Data processing

In Experiment 1, an Exploratory Factor Analysis (EFA) with maximum likelihood extraction and direct oblimin rotation was carried out to examine the associations among the 11 judged attributes (accuracy, authenticity, competence, genuineness, sincerity, honesty, truthfulness, warmth,

likeability, prestige, and considerateness) of Mr. Smith. The Kaiser-Olkin Measure of Sampling adequacy (0.620) and Bartlett's test of sphericity [$\chi^2(3) = 379.19$, $p<0.001$] indicated that the 11 items had adequate common variance for EFA. The analysis revealed that the data had a two-factor structure consisting of factors "Perceived honesty" which explained 74.4% of variance, and "Likeability", which explained 25.6%. Two items (prestige and considerateness) were dropped as they did not significantly contribute to either factor. A two-factor structure was accepted for the remainder of the analyses. In all experiments, the attribute ratings were converted into two scores. Perceived honesty scores were calculated by multiplying each participant's ratings of accuracy, authenticity, competence, genuineness, sincerity, honesty, and truthfulness by their loadings (range 0.61 to 0.98) and dividing by the sum of the loadings. Likeability scores were computed analogously by considering likeability and warmth (factor loadings 0.61 and 0.87, respectively). For Experiments 2, 3, and 4, separate confirmatory factor analyses (available in Section 2 of the online supplement) were carried out to obtain factor loadings for computation of the factor scores for perceived honesty and likeability.

In all experiments, tolerance of norm violations was calculated by forming the mean across items separately for dishonesty, incitement and

expansion of power. All raw data and analysis scripts are available at https://osf.io/tkr56/.

## Assumptions for statistical tests

In Experiment 1, Levene's test for homogeneity of variance was not violated (all $p$s > 0.05) and the data was normally distributed making it suitable for the planned analysis of (co)variance (AN(C)OVA) analyses. However, in Experiment 2, we found Levene's to be violated for the two dependent variables of incitement and dishonesty ($F_{(7286)} = 2.463$, $p = 0.018$ and $F_{(7286)} = 2.253$, $p = 0.030$, respectively) indicating a violation of the homogeneity of variance assumption. While the data was normally distributed for four of the five dependent variables, for dishonesty the Q-Q plot indicated a slight positive skew and upon further investigation revealed skewness = 1.135 and leptokurtosis (kurtosis = 3.912).

Likewise, in Experiment 3, Levene's was violated for the two dependent variables of incitement and dishonesty ($F_{(7441)} = 4.568$, $p<0.001$ and $F_{(7286)} = 2.813$, $p = 0.007$, respectively) indicating a violation of the homogeneity of variance assumption. The Q-Q plot again indicated skewness for dishonesty with a slight positive skew (skewness = 1.312) and was shown to be leptokurtic (kurtosis = 4.394).

In Experiment 4, Levene's was found to be violated for the two dependent variables of Dishonesty and Likeability ($F_{(7550)} = 2.527$, $p = 0.0145$ and $F_{(7550)} = 2.511$, $p = 0.015$, respectively). Again, Dishonesty had a slight positive skew indicated in the Q-Q plot and upon further examination we observed skewness = 1.183 and leptokurtosis (kurtosis = 3.808).

Overall, while there were minor violations of homogeneity of variance, such violations are not uncommon with Likert-scale data and linear model testing is generally robust to these effects (refs. 51–53; cf. ref. 54). Similarly with the minor deviations of normality observed, given the overall sample sizes, distribution per experimental category and the distributions observed in the Q-Q plots, these are unlikely to be a problem for the planned analyses[55]. Nonetheless, to address these violations, in Section 3 of the online supplement we present Welch's f-test results that are robust to such violations[54], which qualitatively confirm the ANOVA results presented below for the dependent variables concerned.

## Results

Figure 2 summarizes the results of all four experiments, with the main effects of each experimental factor shown in their respective columns for all measures across experiments. In each column the direction of the triangle indicates the direction of the effect and its shading the effect size. Significant two-way interactions (labelled $a$ through $f$) are indicated by the arrows connecting the experimental variables involved and are shown in Fig. 3. Although most of our statistical tests involved preregistered hypotheses, which helps protect against the inflation of Type I errors associated with multiple statistical tests[56], we also applied Benjamini-Hochberg (BH) corrections to control for false discovery rates that may arise from multiple comparisons[56,57]. For tractability, we chose to apply the BH corrections for all effects, irrespective of whether they were preregistered or not.

The results are presented by experimental variable, starting with perspective. Here we report the tests that remained statistically significant following the BH corrections. The complete AN(C)OVA tables (including uncorrected $p$-values and the BH corrected $\alpha$-values) and the underlying cell means for all dependent variables are available in Appendix C (ANOVA) and D (ANCOVA). The online supplement contains the Intention to Treat ANOVA tables in Section 6. We also additionally applied a Bonferroni correction to the preregistered ensemble of tests for each experiment. The Bonferroni correction did not affect the outcome of the preregistered hypothesis tests.

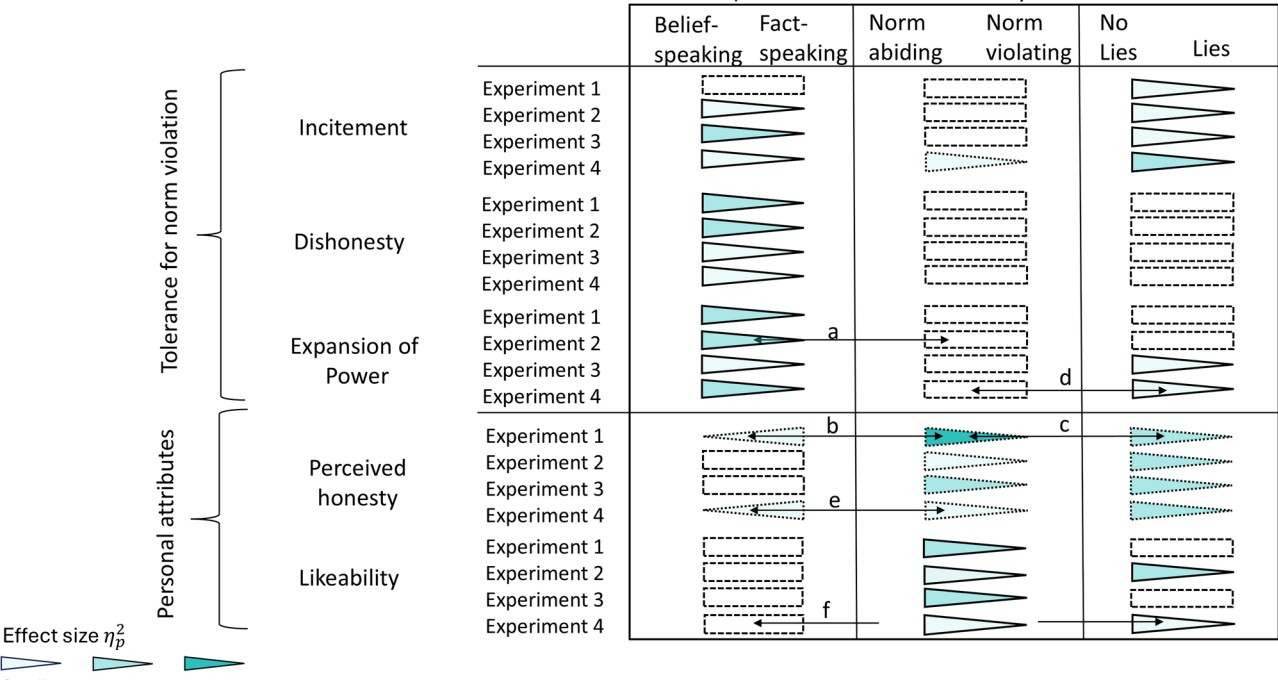

**Fig. 2 | Visualisation of the Benjamini-Hochberg corrected results of all four experiments and the five dependent variables.** For significant main effects, the triangle in the corresponding column is pointing towards the lesser value. For example, in Experiment 1 belief-speaking made participants more tolerant of dishonesty. The dotted outline indicates results for which there were no preregistered hypotheses. Dashed empty rectangles represent null effects. Arrows represent significant interactions between variables and are labelled by letters which refer to panels in Figure 3 that present the interactions. In addition to these, there was one three-way interaction on perceived honesty in Experiment 4. Shading represents effect sizes as per $\eta^2_p$, with light colour indicating small effect size ($\eta^2_p \leq 0.01$), medium colour indicating medium effect size ($0.05 \leq \eta^2_p < 0.14$), and dark colour indicating large effect size ($\eta^2_p \geq 0.14$). For example, in Experiment 1, the effect of the normativity variable on perceived honesty was large, with people rating honesty significantly higher when the politician ("Mr. Smith") was norm abiding than when he violated norms.

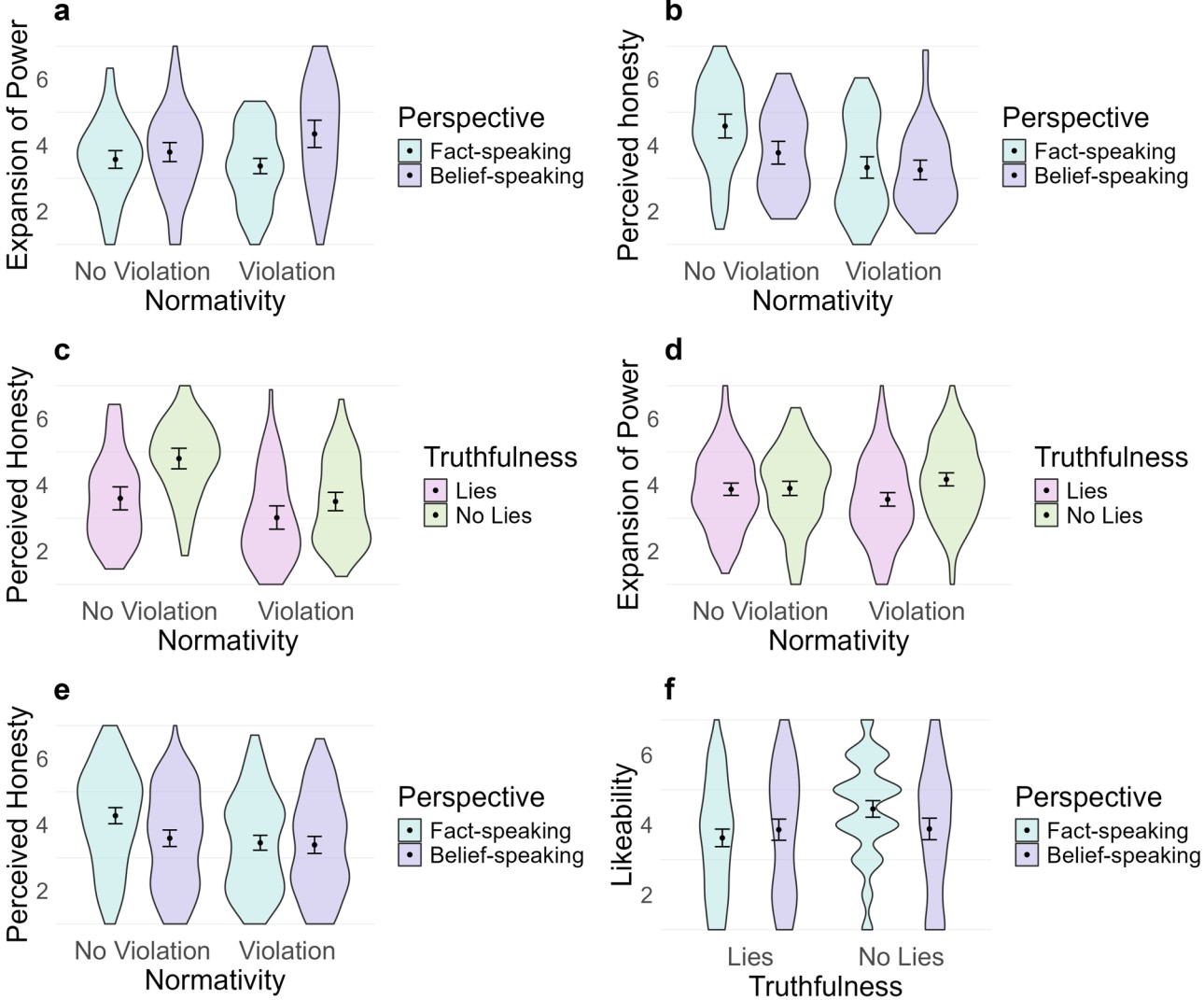

**Fig. 3 | Interactions from the four experiments, means and 95% CIs are illustrated.** Pale turquoise refers to fact-speaking perspective, pastel purple on belief-speaking perspective. Pastel pink refers to existence of lies, whereas pale lime green refers to no lies. Panel (**a**) represents the interaction between perspective and normativity on expansion of power from Experiment 2 (*n* = 294). Panel (**b**) illustrates the interaction between perspective and normativity on perceived honesty, whereas panel (**c**) shows the interaction between normativity and truthfulness on perceived honesty, both from Experiment 1 (*n* = 238). Panel (**d**) represents the interaction between normativity and truthfulness on expansion of power and panel (**e**) represents the interaction between perspective and normativity on perceived honesty, both from Experiment 4 (*n* = 552). Panel (**f**) represents the interaction between perspective and truthfulness on likeability from Experiment 4 (*n* = 552).

## Effects of perspective

When participants took a belief-speaking perspective, they generally became more tolerant of norm violations than when they adopted a fact-speaking perspective. Turning first to incitement, Experiments 2, 3 and 4 (but not 1) showed a small and medium effect of perspective, ($F(1, 286)$ = 12.650, $p<0.001$, $\eta_p^2 = 0.04$, 95%$CI[0.011.00]$; $F(1441)$ = 21.576, $p<0.001$, $\eta_p^2 = 0.05$, 95%$CI[0.02,1.00]$ and $F(1550)$ = 28.747, $p<0.001$, $\eta_p^2 = 0.05$, 95%$CI[0.02,1.00]$, respectively. In all of these cases, belief-speaking raised tolerance for incitement, supporting H3 and H4 (see Table 1).

Similarly, belief-speaking increased tolerance for dishonesty across all four studies. In Experiments 1 and 2, the effect was medium-sized, $F(1230)$ = 21.120, $p<0.001$, $\eta_p^2 = 0.08$, 95%$CI[0.04,1.00]$ and $F(1286)$ = 14.033, $p<0.001$, $\eta_p^2 = 0.05$, 95%$CI[0.01,1.00]$, respectively. In Experiments 3 and 4, there was a small main effect of perspective $F(1441)$ = 11.898, $p<0.001$, $\eta_p^2 = 0.03$, 95%$CI[0.01,1.00]$ and $F(1550)$ = 19.626, $p<0.001$, $\eta_p^2 = 0.03$, 95%$CI[0.01,1.00]$, respectively, supporting hypotheses H3 through H7.

Finally, belief-speaking also made people more accepting of an expansion of power in all four experiments. In Experiment 1, perspective had a medium-sized main effect; $F(1230)$ = 20.099, $p<0.001$, $\eta_p^2 = 0.08$, 95% $CI[0.03,1.00]$. In Experiments 2, 3, and 4, the effect was small, $F(1286)$ = 16.022, $p<0.001$, $\eta_p^2 = 0.05$, 95%$CI[0.02,1.00]$; $F(1441)$ = 21.750, $p<0.001$, $\eta_p^2 = 0.05$, 95%$CI[0.02,1.00]$, and $F(1550)$ = 39.160, $p<0.001$, $\eta_p^2 = 0.05$, 95%$CI[0.02,1.00]$, respectively. These results support our hypotheses H3 through H7 as presented in Table 1.

In addition to the main effects on expansion of power, in Experiment 2 perspective interacted with normativity, $F(1286)$ = 4.745, $p = 0.030$, $\eta_p^2 = 0.02$, 95%$CI[0.00,1.00]$, reflecting the fact that when Mr. Smith was violating norms and participants' perspective was belief-speaking, they were more tolerant of expansion of power than when the behaviour was norm abiding. This interaction is illustrated in panel a of Fig. 3.

Further to the hypothesised main effects just presented, perspective had a small, significant impact on perceived honesty—a factor score consisting of ratings for accuracy, authenticity, competence, genuineness, sincerity, honesty, and truthfulness—in Experiments 1; $F(1230)$ = 7.814, $p = 0.006$, $\eta_p^2 = 0.03$, 95%$CI[0.01,1.00]$ and 4; $F(1550)$ = 11.241, $p = <0.001$, $\eta_p^2 = 0.02$, 95%$CI[0.01,1.00]$, indicating that when participants adopted the fact-speaking perspective, they viewed Mr. Smith as more honest.

In addition, there was a small interaction in Experiment 4 between perspective and normativity on perceived honesty, $F(1550) = 6.074$, $p = 0.014$, $\eta_p^2 = 0.01$, $95\%CI[0.00,1.00]$ (illustrated in panel e of Fig. 3), as well as a small three-way interaction Perspective × Normativity × Truthfulness $F(1550) = 16.51$, $p<0.001$, $\eta_p^2 = 0.02$, $95\%CI[0.00,1.00]$.

There were no statistically significant effects of perspective on likeability in any of the experiments. However, Experiment 4 yielded a small interaction between perspective and truthfulness, $F(1550) = 4.079$, $p = 0.04$, $\eta_p^2 < 0.01$, $95\%CI[0.00,1.00]$, which is illustrated in panel f of Fig. 3. This indicated that when asked to take the perspective of fact-speaking, people generally perceived Mr. Smith as more likeable. However, when Mr. Smith was lying, having the perspective of fact-speaking made Mr. Smith less likeable.

### Effects of normativity

Normativity had no significant effects on tolerance for incitement, dishonesty, or expansion of power in Experiments 1, 2, or 3, whereas in Experiment 4 there was a small effect of normativity on incitement, $F(1550) = 5.022$, $p = 0.025$, $\eta_p^2 < 0.01$, $95\%CI[0.00,1.00]$. Overall, the level of endorsement of norm violations was generally not affected by whether the politician had already violated norms in the scenario.

However, perceived honesty of Mr. Smith was affected by normativity in all studies. In Experiment 1 there was a large main effect, $F(1230) = 36.194$, $p<0.001$, $\eta_p^2 = 0.14$, $95\%CI[0.07,1.00]$, in Experiments 2 and 4 there was a small effect $F(1286) = 9.247$, $p = 0.003$, $\eta_p^2 = 0.03$, $95\%CI[0.01,1.00]$, $F(1550) = 22.698$, $p<0.001$, $\eta_p^2 = 0.04$, $95\%CI[0.00,1.00]$, and in Experiment 3 it was medium-sized $F(1441) = 44.669$, $p<0.001$, $\eta_p^2 = 0.09$, $95\%CI[0.05,1.00]$. This indicates that when Mr. Smith was norm abiding, he was perceived as more honest than when he was violating norms. Further to this, as illustrated in panel b of Fig. 3, in Experiment 1 there was an interaction between perspective and normativity $F(1230) = 5.637$, $p = 0.018$, $\eta_p^2 = 0.008$, $95\%CI[0.00,1.00]$, as well as between normativity and truthfulness $F(1230) = 4.982$, $p = 0.026$, $\eta_p^2 = 0.02$, $95\%CI[0.00,1.00]$ (panel c of Fig. 3) on perceived honesty. These interactions indicate that when a participant's perspective was fact-speaking and Mr. Smith was violating norms, his attempts at expanding his power were less tolerated than if a participants' perspective was that of belief-speaking. Further, when Mr. Smith was lying and violating norms, he was perceived as less honest than when he was not lying and behaved normatively.

In all four experiments there was a main effect of normativity on the factor score of likeability, which consisted of attributes likeability and warmth. In the first three experiments it was medium-sized: Experiment 1 $F(1230) = 18.689$, $p<0.001$, $\eta_p^2 = 0.08$, $95\%CI[0.03,1.00]$, Experiment 2 $F(1286) = 27.844$, $p<0.001$, $\eta_p^2 = 0.09$, $95\%CI[0.04,1.00]$, Experiment 3 $F(1441) = 26.729$, $p<0.001$, $\eta_p^2 = 0.06$, $95\%CI[0.03,1.00]$, and in Experiment 4 it was small $F(1550) = 22.619$, $p<0.001$, $\eta_p^2 = 0.04$, $95\%CI[0.02,1.00]$. When Mr. Smith was violating the norms of incitement, dishonesty, and expansion of power, he was viewed as less likeable than when his behaviour was normative, supporting H5 as illustrated in Table 1. After the BH corrections there were no statistically significant effects, supporting H6 in Table 1.

### Effects of truthfulness

Turning to truthfulness, there was a small main effect of lies on tolerance of incitement in the first three experiments; Experiment 1 $F(1230) = 8.923$, $p = 0.003$, $\eta_p^2 = 0.04$, $95\%CI[0.01,1.00]$, Experiment 2 $F(1286) = 15.729$, $p<0.001$, $\eta_p^2 = 0.05$, $95\%CI[0.02,1.00]$, and Experiment 3 $F(1441) = 6.598$, $p = 0.01$, $\eta_p^2 = 0.01$, $95\%CI[0.00,1.00]$. In Experiment 4, the effect was medium-sized, $F(1550) = 35.151$, $p<0.001$, $\eta_p^2 = 0.06$, $95\%CI[0.06,1.00]$. This consistently shows that when Mr. Smith is lying, participants are less tolerant of his attempts to incite the public.

For dishonesty, there were no significant main effects of truthfulness in Experiments 1, 2, 3, or 4. However, truthfulness had a significant, small, main effect on expansion of power in Experiment 4 $F(1550) = 10.320$,

$p = 0.001$, $\eta_p^2 = 0.02$, $95\%CI[0.00,1.00]$, indicating that when Mr. Smith was not lying, his attempts at expansion of power were more tolerable.

Truthfulness also had a significant, medium-sized, effect on the perceived honesty factor; Experiment 1 $F(1230) = 20.487$, $p<0.001$, $\eta_p^2 = 0.08$, $95\%CI[0.03,1.00]$, Experiment 2 $F(1286) = 22.490$, $p<0.001$, $\eta_p^2 = 0.07$, $95\%CI[0.03,1.00]$, Experiment 3 $F(1441) = 27.682$, $p<0.001$, $\eta_p^2 = 0.06$, $95\%CI[0.03,1.00]$, and Experiment 4 $F(1550) = 58.482$, $p<0.001$, $\eta_p^2 = 0.10$, $95\%CI[0.06,1.00]$. Thus, when Mr. Smith was lying, he was consistently perceived as less honest.

For the likeability factor in Experiments 2 and 4, there was a small main effect of lies $F(1286) = 9.726$, $p = 0.002$, $\eta_p^2 = 0.03$, $95\%CI[0.01,1.00]$; $F(1550) = 9.355$, $p = 0.002$, $\eta_p^2 = 0.02$, $95\%CI[0.00,1.00]$, respectively, indicating that when Mr. Smith was lying, he was perceived as less likeable.

Overall, across the four experiments we discovered that six out of the nine preregistered hypotheses presented in Table 1 were supported. We also obtained consistent evidence for the role of normativity and truthfulness, although most of these effects were not preregistered. However, the most prominent and consistent result is that, as hypothesised and preregistered, when people are directed to rely on sincerity rather than accuracy as the main criterion of honesty, they became more tolerant of democratic norm violations.

### Role of political orientation

In Experiment 4, we measured political orientation using an 11-point scale (where higher values reflect more right-wing orientation). We first examined the association between political orientation and E2IS scores and observed a small, negative relationship, $r = -0.22$, $p<0.001$, $95\%CI[-0.30,-0.14]$. Thus, when a person prefers intuition over evidence (i.e., scores lower on the E2IS), they are more likely to be on the right of the political spectrum, supporting H9 from Table 1. This is also in line with previous work illustrating the preference of conservatives for intuition over evidence[11,23].

We next examined how political orientation affected our experimental interventions. As participant recruitment on Prolific was possible based on their pre-declared political orientation ("left" or "right"), these were used as categories for analysis. Figure 4 compares pre-declared and experimentally measured political orientation and shows that these largely corresponded. Moreover, a nested model comparison of a four-way ANOVA (Perspective × Normativity × Truthfulness × Political Orientation), and a three-way ANCOVA with political orientation instead used as a covariate (replacing the left-right factor) for all of the five dependent variables (incitement, dishonesty, expansion of power, perceived honesty and likeability) yielded statistically non-significant results, all $p$'s > 0.05, implying that use of the continuous measure did not explain any additional variance not already available from the pre-declared binary variable. Thus, it was possible to utilise the pre-declared political orientation (left or right) as the covariate for analyses.

Political orientation was a significant covariate—with effect-sizes varying from small to medium—for four of the five dependent variables of incitement $F(1549) = 42.29$, $p<0.001$, $\eta_p^2 < 0.01$, $95\%CI[0.00,1.00]$, expansion of power $F(1549) = 11.145$, $p<0.001$, $\eta_p^2 = 0.02$, $95\%CI[0.01,1.00]$, perceived honesty $F(1549) = 50.013$, $p<0.001$, $\eta_p^2 = 0.08$, $95\%CI[0.05,1.00]$, and likeability $F(1549) = 65.514$, $p<0.001$, $\eta_p^2 = 0.11$, $95\%CI[0.07,1.00]$, supporting H8 in Table 1. It was not a significant covariate for dishonesty. Figure 5 illustrates these main effects across all dependent variables. Full ANCOVAs are reported in Appendix D.

### Exploratory analyses

**Effects of intervention on right-wing participants.** As we just showed, people with a right-wing orientation typically prefer an intuition-based approach to truth and are also generally more tolerant of norm violations. It is therefore of interest to examine whether our experimental interventions, in particular the assignment of different perspectives of honesty, can also be observed with the right-wing participants in Experiment

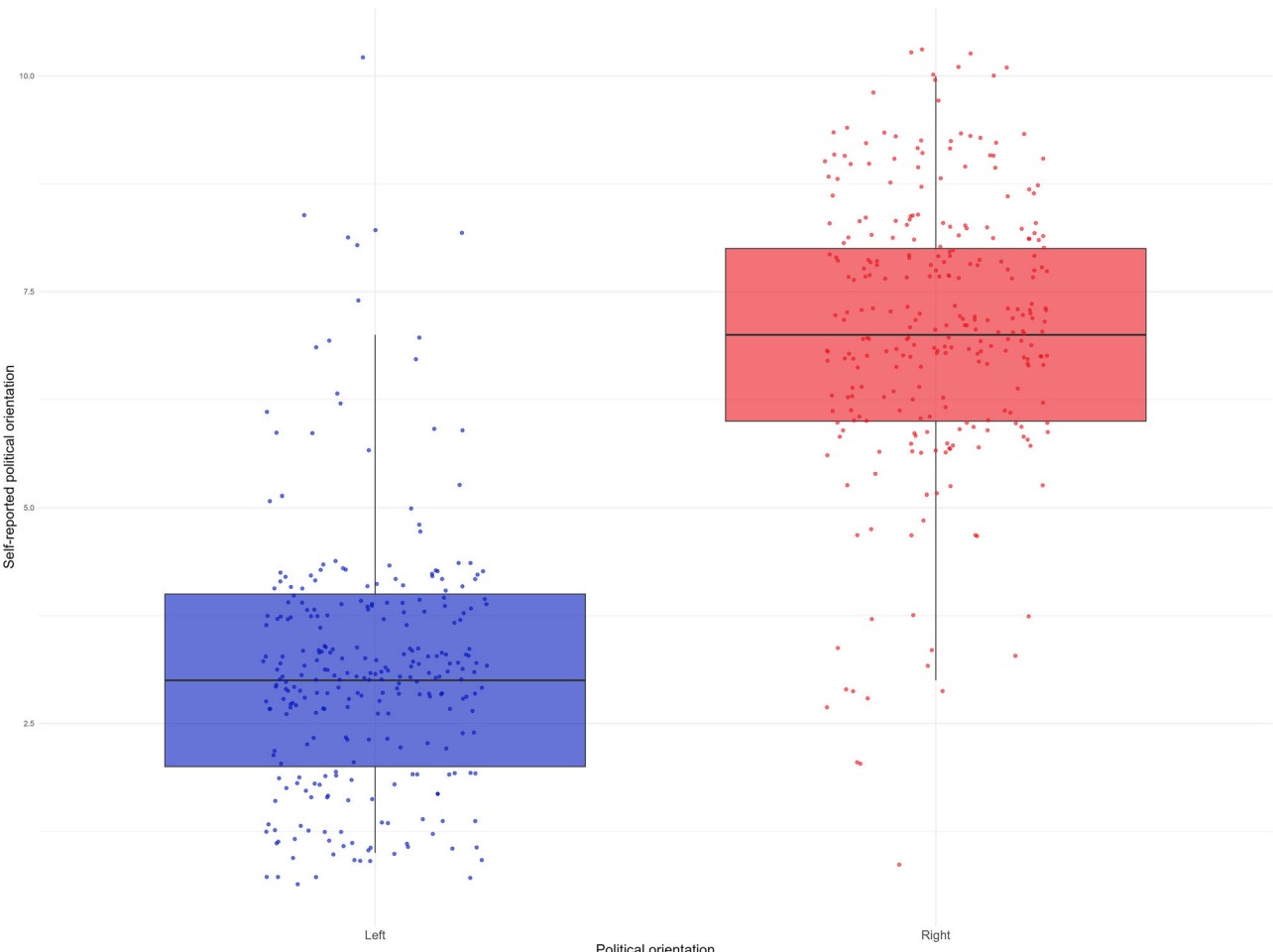

**Fig. 4 | Pre-declared (left or right) and self-reported (11-point scale where higher numbers indicate orientation closer to the right) political orientation in Experiment 4.** The line represents the median, the bottom the 25th percentile and the top the 75th percentile. Total $n = 552$.

4. We conducted a Perspective × Normativity × Truthfulness ANOVA on the subset of self-declared right-wing supporters ($n = 263$). Below we report BH corrected results.

Turning first to perspective, there was a small main effect on expansion of power $F(1263) = 10.800$, $p = 0.001$, $\eta_p^2 = 0.04$, 95%$CI[0.01,1.00]$, and perceived honesty $F(1263) = 7.181$, $p = 0.008$, $\eta_p^2 = 0.03$, 95%$CI[0.00,1.00]$. There were no statistically significant main effects on incitement, dishonesty or likeability.

Looking at normativity, there was a medium-sized main effect on perceived honesty $F(1263) = 17.500$, $p<0.001$, $\eta_p^2 = 0.06$, 95%$CI[0.02,1.00]$, and a small interaction between normativity and perspective $F(1263) = 6.203$, $p = 0.013$, $\eta_p^2 < 0.01$, 95%$CI[0.00,1.00]$. There was also a small main effect of normativity on likeability, $F(1263) = 10.604$, $p = 0.001$, $\eta_p^2 = 0.04$, 95%$CI[0.01,1.00]$.

Finally, looking at truthfulness, there was a medium-sized main effect on incitement $F(1263) = 19.804$, $p<0.001$, $\eta_p^2 = 0.07$, 95%$CI[0.07,1.00]$, a medium-sized main effect on perceived honesty $F(1263) = 40.973$, $p<0.001$, $\eta_p^2 = 0.013$, 95%$CI[0.08,1.00]$, and a small main effect on likeability $F(1263) = 10.756$, $p = 0.001$, $\eta_p^2 = 0.04$, 95%$CI[0.01,1.00]$. In addition to these, there were two small interactions of truthfulness and perspective on likeability $F(1263) = 12.690$, $p = 0.014$, $\eta_p^2 = 0.02$, 95%$CI[0.00,1.00]$, and truthfulness and normativity expansion of power $F(1263) = 8.036$, $p = 0.005$, $\eta_p^2 = 0.03$, 95%$CI[0.00,1.00]$. Full ANOVA tables for these are reported in Appendix C.

These results show that our intervention worked even on those who are on the right of the political spectrum, and who are therefore most likely to favour an intuition-based perspective by default. While we see

that the magnitude of the effect of perspective on participants on the right is smaller than it is on those on the left, the effect still exists. Considering the means across the three norm violations of incitement, dishonesty, and expansion of power, we find that for people on the left, assuming the fact-speaking perspective (as opposed to belief-speaking) reduced their tolerance for norm violations by 0.63 (on a 7-point scale), whereas for people on the right the reduction was a more modest 0.35. In absolute terms, when people on the right are asked to take a fact-speaking perspective ($M = 3.33$), they are slightly less tolerant than people on the left who are asked to take a belief-speaking perspective ($M = 3.44$) but remain more tolerant than people on the left who are asked to take a fact-speaking perspective ($M = 2.80$). Overall, it is encouraging that right-wing supporters' tolerance for norm violations can be significantly reduced by a relatively simple intervention.

**Role of epistemic beliefs**. To examine the impact of a participants' pre-existing preference for evidence versus intuition, we repeated the main analyses using ANCOVAs with the Epistemic Evidence Intuition Scale (E2IS; ref. 50) scores as covariates. Experiment 1 was excluded from these analyses as the full E2IS data was not available for those participants, only their scores for accuracy and sincerity. Here we only report results that differ statistically from those without the covariate. Full results of the ANCOVAs can be found in Appendix D of this manuscript. Further information about E2IS can be found in Section 7 of the online supplement.

In Experiment 2, inclusion of the covariate led to the emergence of a small interaction between truthfulness and perspective on dishonesty

**Fig. 5 | All main effects from Experiment 4, means and 95% CIs are illustrated.** Panel (**a**) illustrates the main effect of political orientation on incitement, Panel (**b**) on dishonesty, and Panel (**c**) on expansion of power. The last two panels showcase the main effects of political orientation on personal attributes of the politician, Panel (**d**) on perceived honesty and Panel (**e**) on likeability. $n = 552$.

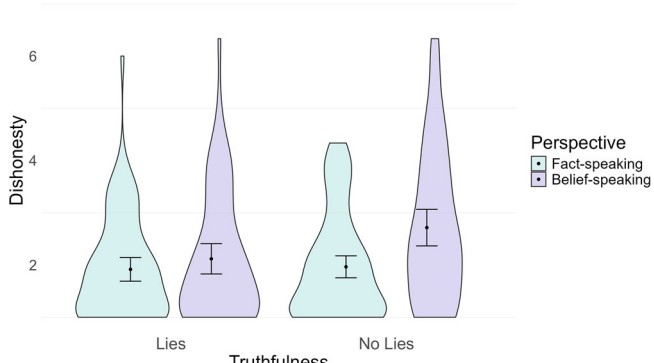

**Fig. 6 | Additional interaction between perspective and truthfulness on dishonesty from Experiment 2 when E2IS scores are included as a covariate.** Mean and 95% CIs are shown. $n = 294$.

$F(1259) = 4.331$, $p = 0.038$, $\eta^2_p = 0.02$, 95%$CI[0.01,1.00]$. The E2IS scores for 26 of the participants are missing, hence the smaller df. Figure 6 illustrates the interaction.

E2IS was a significant covariate for expansion of power in Experiments 2 $F(1259) = 7.656$, $p = 0.006$, $\eta^2_p = 0.03$, 95%$CI[0.00,1.00]$, 3; $F(1440) = 21.961$, $p<0.001$, $\eta^2_p = 0.01$, 95%$CI[0.00,1.00]$, and 4; $F(1549) = 10.623$, $p = 0.001$, $\eta^2_p = 0.02$, 95%$CI[0.00,1.00]$.

In Experiment 3, E2IS was also a significant covariate for perceived honesty $F(1440) = 4.255$, $p = 0.04$, $\eta^2_p = 0.01$, 95%$CI[0.00,1.00]$.

In Experiment 4, E2IS was a significant covariate for incitement, $F(1549) = 10.495$, $p = 0.001$, $\eta^2_p = 0.02$, 95%$CI[0.03,1.00]$ as well as for dishonesty, $F(1549) = 13.668$, $p<0.001$, $\eta^2_p = 0.02$, 95%$CI[0.01,1.00]$.

Overall, higher scores on the E2IS indicated preference for evidence over intuition. We find that participants who scored higher on the E2IS were less likely to tolerate expansion of power or perceive Mr. Smith as honest than those who scored lower and preferred intuition over evidence.

All in all, across the four experiments we obtained consistent evidence that when people are asked to perceive sincerity over accuracy as the main criterion of honesty, they became more tolerant of democratic norm violations. This effect is present regardless of person's political orientation or pre-existing epistemic preferences.

## Discussion

In all four experiments, we consistently found that when participants were asked to take a belief-speaking perspective—that is, when they were asked to view sincerity and authenticity as central components of honesty—people were more likely to tolerate norm violations by political elites. We also found that norm violations have a significant impact on the perceived honesty and likeability of a fictitious political leader; when he is violating norms, he is seen as less honest and likeable. In addition, the identification of specific lies in the scenarios resulted in the politician being viewed as less honest and also reduced tolerance for norm violations.

Although our intervention did not aim to fundamentally change a participants' beliefs, the results suggest that our intervention was arguably more powerful than a participant's inherent preference for evidence over intuition. Although E2IS scores were a significant covariate for the analyses involving expansion of power and perceived honesty in Experiments 2 and 3, the E2IS scores played no significant role in the other dependent variables. This suggests that although pre-existing conceptions of honesty can influence people's assessment of a political leader's behaviour, in our data this impact was small in comparison to the impact of experimentally directed perspective-taking. By contrast, in Experiment 4, participants' pre-existing political orientation was a significant covariate for all of our dependent variables (incitement, dishonesty, expansion of power, perceived honesty, and likeability). However, crucially, neither epistemic preferences nor political orientation overrode our experimental intervention: the main effect of perspective remained significant in the ANCOVA models, and assigned perspective affected tolerance for norm violations even in the subset of right-wing participants in Experiment 4. We conclude that interventions that shift people towards an accuracy-based notion of honesty can reliably reduce tolerance for democratic norm violations by political elites across the political spectrum. We next take up some potential limitations of our study before we place our results into a broader theoretical context.

## Limitations

In general, there is evidence that perspective-taking even on controversial issues—such as Syrian refugees or in situations of inter-group conflict—can successfully induce behaviour in the short-term[58–60]. Our results add to that body of work. Our main experimental intervention sought to direct people towards adopting one or the other component of honesty, sincerity or factual accuracy, for the duration of the experiment. While this was demonstrably successful in all experiments, and hence of considerable practical import, it is important to delineate the psychological consequences of our intervention.

It is unlikely, and we do not claim, that our intervention altered participants' fundamentally held beliefs about honesty. This is also unnecessary: our assumption was that people generally are conversant with both aspects of honesty and are therefore readily able to prefer one or the other when directed to do so by the intervention. There is support for that assumption both within our data and in the existing literature.

First, the vignettes used here have been successfully used in previous research where people were asked to pick which one they prefer[46]. The earlier work demonstrated that people are conversant with the two components of honesty because their preferences for one or the other vignette correlated with the same indicators of epistemic preferences that we used in our Experiment 1[46]. In further support, it has been shown that people adjust to fact-speaking or belief-speaking rhetoric simply by being asked to respond to text that is predominantly accuracy- or sincerity-based[61]. Similarly, when asked to judge which component of honesty is prevalent in a given text, people's judgments are closely aligned (AUC ≃ 0.80) with the results of a computational linguistic analysis using prevalidated dictionaries[44,61]. People can also readily generate times from their lives when they were authentic or sincere but did not tell the truth[62], further attesting to people's familiarity with those two aspects of honesty. Moreover, we reported at the outset how identifiable lies can become acceptable when cloaked as authenticity[33], precisely as expected if people are conversant with those two forms of honesty. Overall, there is little doubt that people are familiar with the distinction between sincerity and factual accuracy and can demonstrably recognize or adapt one or the other perspective on honesty in a variety of tasks and in response to a variety of cues.

Second, we observed an effect of the intervention even when statistically controlling for people's prior epistemic preferences. While epistemic preferences were a significant covariate for four of the dependent variables of incitement (Experiment 4), dishonesty (Experiment 4), expansion of power (Experiments 2, 3, and 4), and perceived honesty (Experiment 3), this effect was not consistent across the three experiments. Expansion of power was the only dependent variable where the covariate was significant in Experiments 2 through 4. Thus, it was not as strong or as significant as the impact of our

intervention, particularly since the intervention remained significant in these cases as well. This renders it unlikely that participants struggled to internalise the assigned perspective when it conflicted with their pre-existing epistemic preferences. On the contrary, our intervention arguably is more powerful than pre-existing epistemic preferences, further attesting to the practical importance of our results.

We next discuss the broader implications of our work by taking up each experimental factor in turn.

## Normativity

Whether or not the politician violated democratic norms in the scenario did not have a statistically significant effect on people's tolerance of norm violations, although it did affect perceived honesty and likeability. There are several possible reasons for this outcome. One possibility is that the specific, individual norm violations considered in isolation are not considered important even though the combined societal effects of them are[16,63]. Thus, people's attitudes towards democratic norms may transcend the specifics of their experiences; they will tolerate hypothetical future violations as much or as little as any actual violations they have already witnessed.

Moreover, in our experiments, the politician's rise to power was not questioned. The politician in our scenarios was presented as a legitimate figure and he claimed that the norm violations would be for a greater good. It is therefore possible that the norm violations were not seen as destabilising the democratic system as a whole. Previous work by Hinterleitner and Sager[64] has shown that non-democratic actions committed by actors who have risen to power through democratic means can be perceived to be democratic and acceptable. This is particularly true for those who supported the politician to begin with. For instance, when Prime Minister Boris Johnson's prorogation of the U.K. Parliament was ruled unlawful by the supreme court, 49% of the general population thought that he should resign but only 22% of Conservative voters (vs. 73% of Labour voters) thought so[65]. Indeed, previous work[33] has established that while people do not like lying demagogues, they may still tolerate them; our results display a similar pattern.

## Truthfulness

We found greater tolerance of norm violations for politicians that were identified as truthful in the scenario. This result may appear to be counterintuitive at first glance. Although dishonesty is not uncommon in modern societies and can be beneficial for political success[66,67], tolerance of dishonest politicians is context dependent[68,69]. On the one hand, the public still generally expects politicians to be honest[70]. Moreover, when honesty is made a salient value, it influences people's assessment of political leaders and makes lies less tolerable[71]. Similarly, it is likely that in our experiments, the perspective intervention made participants focus on the honesty of the politician, so when that politician was shown to be truthful, he may have benefited from greater largesse.

On the other hand, when people are not explicitly asked to focus on honesty, previous work has shown that lies may not significantly influence politically relevant variables, such as voting intentions[25]. The existence of such divergent outcomes raises important questions about the role of honesty in political judgements and underscores the need to consider the context when seeking to understand how people evaluate the honesty of politicians.

## Perspective

Our key finding, namely that people can be directed to take different perspectives on honesty and that this affects their tolerance for norm violations, is particularly relevant in light of the recent globally increasing preponderance of extreme-right views under the banner of populism. Previous research has identified belief-speaking[44,72] and democratic norm violations[73] as central aspects of right-wing populism.

Right-wing populists' regular appeals to people's "common sense"[20] can be powerful when it leads people to prefer sincerity over accuracy. Indeed, populist politicians have been characterised as displaying a

particular set of behaviours, which ultimately makes them more charismatic[74]. Young et al.[20] have shown that various populist attitudes, such as valuing intuition over evidence and belief in misinformation, occur concurrently and can be mutually reinforcing. It is therefore particularly encouraging that our fact-speaking intervention reduced people's willingness to accept politicians' violation of democratic norms. Our work mirrors related experiments by Shayegh, Baysu, and Turner[75], which employed a social-identity-based intervention to influence populist attitudes and anti-immigration attitudes. Experimental interventions go beyond most existing research because much of the work on democratic norms has focussed on correlational rather than experimental work [for an extensive review see e.g., ref. 23].

One possible mechanism by which our perspective intervention may have shifted people's tolerance for norm violations is that when we encouraged acceptance of belief-speaking, we activated a cluster of populist attitudes which primed a person to tolerate norm violations as they are part of the same attitude cluster. Conversely, encouraging fact-speaking may have entailed suppression of the entire populism cluster, thereby also rendering tolerance for norm violations less likely. It remains for future research to investigate that possibility or other alternatives. For now, the exact mechanism underlying our effect is not particularly relevant from a pragmatic point of view—what matters is that there is a tight linkage between notions of honesty and the acceptability of norm violations by politicians, and that directing people towards a more accuracy-focused conception of honesty can make them less accepting of anti-democratic behaviours.

Recent analyses of democratic backsliding have identified elite norm violations to be a crucial underlying contributor[29,76]. Elite norm violations can occur on a variety of levels, ranging from persistent dishonesty (such as Donald Trump making over 30,000 false or misleading claims during his first term as president[77]) or political elites sharing misinformation online[78], to political manoeuvrings that breach convention or represent a grab for power. Even though democracies have institutional mechanisms that are supposed to curtail elite norm violations, these often work retroactively and can be slow. For instance, when the suspension of the British parliament by Boris Johnson was ultimately found to be unlawful by the U.K. Supreme Court, it had no notable political consequences for Johnson and most of the media continued to support him[17].

Our results show increased tolerance for such behaviours in people who prefer (or are asked to prefer) belief-speaking over fact-speaking. This finding has significant implications for modern democracies. It suggests that politicians who pursue an authoritarian agenda can successfully market their authenticity and sincerity and get a free pass for undermining democracy, even if they may be considered to be less honest as a result. Because dishonesty frequently does not seem to carry an electoral price tag [e.g., refs. 25,27,28], authoritarian politicians may accept a reduction in perceived honesty as a small price to pay for the public's endorsement of their efforts at undermining democracy.

## Data availability

All raw data and materials used in the experiments and analyses are openly available at https://osf.io/tkr56.

## Code availability

All analysis codes are openly available at https://osf.io/tkr56. Data were analysed with R version 4.2.2 (2022-10-31 ucrt)[79].

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

## Acknowledgements

The authors have received funding from the European Research Council (ERC) under the European Union's Horizon 2020 research and innovation programme under grant agreement No 101020961 (PRODEMINFO) to Stephan Lewandowsky. Stephan Lewandowsky also acknowledges support from the Humboldt Foundation through a research award and SL's participation in Horizon Europe Project SoMe4Dem is supported by UKRI grant number 10049415 (University of Bristol). The funders had no role in study design, data collection and analysis, decision to publish or preparation of the manuscript. The authors would like to thank Christoph M. Abels for his assistance in elaborating on the construction and validation of the Epistemic-Evidence Intuition-scale.

## Author contributions

Both authors contributed to conceptualisation and methodology. Kiia Jasmin Alexandra Huttunen was in charge of data collection, formal analysis and writing the original draft as well as reviewing and editing subsequent versions. Stephan Lewandowsky contributed to reviewing and editing manuscript drafts, provided supervision, and acquired funding for the project.

## Competing interests

The authors declare no competing interests.
