## [Transparent Peer Review file · Communications Psychology]

Tolerance for democratic norm violations increases when sincerity replaces accuracy as a marker of honesty

Corresponding Author: Ms Kiia Huttunen

Version 0:

Decision Letter:

Dear Ms Huttunen,

Thank you for your patience during the peer-review process. Your manuscript titled "The evolution of truth in political discourse from fact to feeling and its implications for democracy" has now been seen by 2 reviewers, whose comments are appended below. I have discussed the reports with my colleagues and I regret to inform you that we decided that in light of the referee reports, we cannot publish your manuscript in Communications Psychology.

You will see that the reviewers, albeit enthusiastic about the research question, raise substantive methodological concerns. Taking these points together with our editorial considerations, these reservations preclude publication of this study in Communications Psychology.

I am sorry that we cannot be more positive on this occasion and thank you for the opportunity to consider your work.

Best regards,

Marike

Marike Schiffer, PhD
Chief Editor
Communications Psychology

REVIEWERS' EXPERTISE:

Reviewer #1 political psychology

Reviewer #2 political psychology

REVIEWER COMMENTS:

Reviewer #1 (Remarks to the Author):

Review of "The evolution of truth in political discourse from fact to feeling...."

I have read this paper with considerable interest and found much to like about it. At the same time, I think that much more needs to be done to illuminate the role of ideological and psychological variables (see below).

The introduction provides a very useful summary and review of social scientific research on how right-wing populists around the world have contributed to an informational climate in which objective considerations such as fact and evidence are devalued and subjective feelings about what is true are prioritized. I was surprised, however, given that this manuscript has been submitted to a psychology journal, that there was really no discussion of theory and research in social, personality, or political psychology. Consequently, no satisfying psychological account of why right-wing populists focus on feelings rather than facts was provided.

Fortunately, this problem is easily remediable, because a lot of research in political psychology has documented left-right

asymmetries in epistemic motives and abilities. In Chapter 7 of a book entitled *Left & Right: The Psychological Significance of a Political Distinction*, Jost (2021, pp. 230-281) reviewed dozens of studies documenting associations between ideology and various measures of epistemic preferences that are very close to the Evidence-Intuition Scale used in the present research.

This may be something dictated by the journal's formatting instructions, but I did not like the fact that the descriptions of research methods, participants, materials, and procedure came at the very end of the paper. I did not find it possible to understand or interpret the results section without finding the methods section first. The transition from the introduction to the results was abrupt, to say the least.

In any case, looking carefully at the methods section, I found that I did not like the exclusion criteria that were apparently pre-registered. The researchers appear to have excluded any participants who stated that they did not adopt the assigned perspective. This is not a common way of dealing with experimental manipulations, which can surely affect people without their full understanding of how or why. What if some participants were affected by the manipulation but thought (or said) they weren't? Is it worth destroying the power and beauty of random assignment to drop participants based on their subjective reports about what they thought they were doing, given that they can be wrong what they did? I don't think so.

As it stands, the researchers dropped 15% of the participants from Study 1, 19.5% from Study 2, and 6.5% from Study 3. I would much rather see analyses based on the full samples in the main text (with these analyses based on smaller samples included in an online supplement as robustness checks), because only then is random assignment preserved and there is greater statistical power to test hypotheses. I fear that pre-registering and sticking rigidly to exclusion criteria based on hunches is doing much more harm than good in experimental social psychology.

I cannot tell from either the meager methods section or the online supplement if the researchers measured left-right political orientation (or ideology), but I hope so. There are several reasons for this, including the fact that the entire introduction is about right-wing populism (and not left-wing populism, which is mentioned only in a footnote), and the fact that there is lots of evidence showing that leftists and rightists differ in epistemic motives and abilities (Jost, 2021).

To me, the results section does not really illuminate anything unless the authors consider the role of ideology. Based on previous research, one would surely expect that conservative-rightists would be more likely to value intuition and less likely to value evidence, compared to liberal-leftists. This would presumably make conservative-rightists more tolerant of dishonesty, more likely to favor "belief-speaking," more accepting of power expansion and norm violations, etc. Demonstrating all of this would be very interesting and useful for understanding political psychology.

As it stands, the results section is not very enlightening and reads more like a series of manipulation checks (e.g., people who were told that Mr. Smith was lying and violating norms perceived him as less honest than people who were told he was not lying or violating norms). than theoretically meaningful results. I would also like to see much more use made of individual (and ideological) differences in EIS, which was mentioned only in passing as a "covariate" on p. 11.

I think that this research program has potential, but only if it can be used to illuminate ideological asymmetries and psychological processes involved in democratic backsliding. Otherwise, it is not clear to me what is being explained here. The fact that 80% of research participants read the study materials carefully enough to draw appropriate conclusions about hypothetical candidates is not, in and of itself, much of a scientific discovery, it seems to me.

Reviewer #2 (Remarks to the Author):

This is a very well-written paper on a timely and important topic. The distinction between fact-speaking and belief-speaking is likely to be important. Nevertheless, the reported studies suffer from methodological problems, which make the paper unsuitable for publication in this journal. The most important problems are listed below:

1. The manipulation

The authors conclude that whether a person takes a "belief-speaking" or "fact-speaking" perspective affects acquiescence to democratic norm violations and politicians' dishonesty. They presented participants with a vignette describing the perspective of "Sam" (for belief-speaking) or "Taylor" (fact-speaking) and asked the participants to adopt this perspective when answering subsequent questions ("Please take the perspective of Sam[Taylor] when answering the question that will be shown on the next page"... "Bearing in mind your perspective of Sam/Taylor, please answer the following questions."). According to the pre-registrations "Participants who indicate that they have not adopted their assigned perspective will also be excluded from analyses." Nevertheless, it is not obvious how the request to adopt another person's perspective was interpreted. It is quite possible that the participants were simply trying to imagine how a person with a belief-speaking or fact-speaking perspective would react to democratic norm violations and dishonesty—or in other words, that the results reflect the participants' implicit theories about people with these perspectives, rather than their own reactions to norm violations and

dishonesty from a different epistemological perspective. The paper does not present any evidence that the actual perspectives of the participants were shifted by the manipulation.

Some of the conclusions go even further: “Our data shows for the first time that this preference for belief-speaking can be experimentally manipulated and has drastic flow-on consequences for the acceptance of democratic norm violations. This finding supports the notion that belief-speaking is a fundamental aspect of right-wing populist communication.” No evidence is presented to back up any of these strong claims.

2. The hypotheses

The hypotheses are not made explicit in the paper. The authors write: “We present a series of preregistered studies that examined the conditions under which people acquiesce to democratic norm violations and politicians’ dishonesty. We find that when participants are asked to take a perspective of honesty that emphasises sincerity over accuracy, which we call ‘belief-speaking’, they are more willing to accept norm violations by politicians than if participants take a perspective that emphasises accuracy as a criterion for honesty, which we call ‘fact-speaking’.” They thereafter present a large number of significance tests of effects based on a complex design without making explicit how these correspond to their hypotheses. Furthermore, the hypotheses that are listed in the pre-registrations are different from what one might guess that they would be based on the paper—they appear to mainly concern effects of individual differences (e.g., “Participants who prefer accuracy over sincerity are more likely to...”)—and some of the significance tests that are reported prior to the exploratory section are completely unrelated to these hypotheses. It is hard to figure out which statistical tests were hypothesis tests and which ones were not.

3. Type I and Type II errors

The authors report a large number of significance tests based on a 2*2*2 between-subjects design without specification of corrections for the number of statistical hypotheses to prevent an inflated risk of Type I errors. We must assume that there were no corrections since the authors verified that they have described all corrections when submitting the paper. Without corrections, the results can be considered exploratory rather than confirmatory, and further studies are needed to replicate them before any conclusions can be drawn. It is obvious that if appropriate corrections were made, many of the results, including almost all interactions, would not be non-significant (many of the p-values are marginally below .05).

The sample sizes are small for a complex design with tests of between-group interactions, leading to an inflated risk of Type II error. No relevant power analysis is reported. It seems that statistical power was not considered. In one of the pre-registrations, the authors write: “We will recruit 300 participants as correlations generally stabilize around 250 and there are eight experimental conditions”. Although this is a relevant consideration in research on individual differences, it is separate from the power issue, and it is not directly applicable to the estimation of parameters in a complex experimental design.

4. Measures

The authors use the Evidence-Intuition scale to operationalize pre-existing views of honesty. Leaving aside weaknesses of this scale aside, there are two problems. 1. The authors appear to collapse the three scales without providing any evidence that these have a substantive general factor. 2. The link between the theoretical conception of preference for accuracy over sincerity and the content of the scales is somewhat tenuous. The scales concern faith in intuitive hunches about facts, belief that evidence and facts are important when telling whether something is true, and belief that truth is determined by those with political power. Although these are related to a preference for accuracy, what they have to do with beliefs about honesty and sincerity is less clear.

The “perceived honesty” score (based on factor analyses) of judgments of Mr Smith covers both attributes such as authenticity, sincerity, and genuineness and attributes such as competence and accuracy. This labeling is somewhat confusing given the theoretical distinction between sincerity and accuracy. Furthermore, what is the reason for the model misfit (reported in a supplement)?

5. Transparency

There are no valid links to pre-registrations, no code availability statement, no clear description of hypotheses or corrections, and no description of the exact sample size for each experimental condition in the paper.

6. Other claims

What evidence is there for this claim: “People’s subjective conceptions of truth and honesty have undergone significant changes in recent decades. Parts of society increasingly favour the sincere expression of personal belief, however inaccurate, over verifiable facts.”?

Note on appeals: In exceptional circumstances, it is in authors’ interest to appeal an editorial decision. More information on appeals is available here: <https://www.nature.com/commpsychol/submit/editorial-process#appeals>

Decision Letter:

Dear Ms Huttunen,

Thank you for your correspondence asking us to reconsider our decision on your Article, "The evolution of truth in political discourse from fact to feeling and its implications for democracy". After careful consideration we have decided that we would be willing to consider a full appeal in the form of a revised version of your manuscript.

Along with your revised manuscript, you should also submit a separate point-by-point response to all of the concerns raised by the referees, in each case describing what changes have been made to the manuscript or, alternatively, if no action has been taken, providing a compelling argument for why that is the case.

Please note that we will only take the appeal forward and contact the reviewers again if we are persuaded that a substantial attempt has been made to address all editorial concerns and referees' comments. In this case, your revised manuscript and the point-by-point reply will be sent back to the referees so that they can judge whether their concerns have been addressed satisfactorily or otherwise.

I should stress, however, that we would be reluctant to trouble our referees again unless we thought that their comments had been addressed in full.

The key editorial concerns were methodological (incl. regarding success of the experimental manipulation, validity of measures, and statistical strength) and related to the advance inherent in the study. For us to be in a position to send your work back to the reviewers, it is of critical importance that the methodological concerns are satisfactorily addressed. In addition, we strongly encourage you to include additional data to address the aspect of ideological orientation (Reviewer #1), as this would contribute to the novel insights over the existing literature that your study may offer.

Although the following issues were not the reason the previous submission was rejected, to prepare your resubmission in the best possible way for editorial evaluation and potential re-review we ask that:

1) You follow our preregistration policy (<https://www.nature.com/commspsychol/editorial-policies/preregistration-policy>).

Applied to the present case this means:

- a) Clearly highlighting which hypotheses were preregistered and which analyses were preregistered (for an analysis to be considered preregistered, the statistical test and outcome measures would need to have been prespecified); clarifying the match between hypotheses (and each analysis) in the paper with its preregistration.
- b) Reporting the preregistered analysis and the alternative analysis requested by the Reviewers, highlighting which ones are preregistered and which ones are exploratory/based on reviewer request

2) You demonstrate that you had sufficient power of 80% (or more) for the smallest effect size of theoretical interest (with a justification for what would constitute the smallest effect size of theoretical interest). Please note that we are not requesting a post-hoc power analysis for the effect sizes established in the reported analysis.

If the revision process takes significantly longer than five months, we will be happy to reconsider your paper at a later date, provided it still presents a significant contribution to the literature at that stage.

Please use the following link to submit your revised manuscript, point-by-point response to the Reviewers' comments with a list of your changes to the manuscript text (which should be in a separate document to any cover letter) and any completed checklist (if you do not collect any new data, you may bring forward the currently provided Reporting Summary and Editorial Policy Checklist):

Link Redacted

Best regards,

Marike

Marike Schiffer, PhD
Chief Editor
Communications Psychology

* TRANSPARENT PEER REVIEW: Communications Psychology uses a transparent peer review system. This means that we publish the editorial decision letters including Reviewers' comments to the authors and the author rebuttal letters online as a supplementary peer review file. However, on author request, confidential information and data can be removed from the published reviewer reports and rebuttal letters prior to publication. If your manuscript has been previously reviewed at another journal, those Reviewers' comments would not form part of the published peer review file.

Version 2:

Decision Letter:

Dear Kiia,

Thank you for your patience during the peer-review process of your appeal. Your revised manuscript titled "Subjective conceptions of honesty and tolerance for democratic norm violations" has now been seen by 2 reviewers, and I include their comments at the end of this message.

The reviewers, who are both returning referees from the first round, disagree in the assessment of your work, including on fundamental issues. Given the severely diverging views on methodological issues, I had an extensive consultation with Prof Hannah Nam, our editorial board member for political psychology. Following a detailed evaluation of all arguments and benchmarking against publications in the field, we decided that while we cannot accept your paper for publication, we would be willing to assess a revised manuscript.

In detail:

Reviewer 2 lists presentational concerns about the hypotheses and analysis. As for the hypotheses, you provided a Figure that explains how the wording in the main text maps onto the wording in the various preregistration documents, with the wording in the main text representing an umbrella summary of two mirrored/twinned hypotheses. Although this summary is efficient and all hypotheses are listed in the Figure, we ask that you increase redundancy to achieve greater transparency. Please report the hypotheses for each experiment in the main text exactly as written in the preregistration. There is no word limit on the Methods section or Results section. Please also ensure you order the manuscript sections appropriately, i.e. with the Methods section between Introduction and Results, which will improve readability. Please ensure that causal claims are appropriately limited to causal manipulation.

As for the concerns regarding demand effects, the revision needs to adopt at a minimum a more careful presentation and a caveated interpretation.

Further, there are still unresolved issues regarding the evidence-intuition scale. These concerns need to be addressed, at the least through a clarification of the scale's measurement properties and conceptual relation to the manipulation constructs.

For the points above (manipulation and scale), we ask that where you can, you address these concerns with data, rather than purely rhetorically.

Your hypotheses were preregistered and you applied corrections for multiple comparisons. In addition, please ensure that your reporting and interpretation of the results, and especially of non-significant findings, follow the journal's standards, as summarized here: <https://www.nature.com/commspsychol/submit/submission-guidelines#statistical-guidelines> and referenced in greater detail in the attached Editorial Requests Table.

Please attend to each item in the Editorial Requests Table and ensure your manuscript is fully compliant. If your revised manuscript is not aligned with these requests on major issues, such as those concerning statistics, it may be returned to you for further revisions without re-review.

Please submit the following items:

- Revised manuscript
- Point-by-point response to the referees' comments
- Cover letter (as a separate document)
- <https://www.nature.com/documents/nr-reporting-summary.pdf> Nature Research Reporting Summary

- Completed Editorial Request Table (attached).

via this link: Link Redacted .

Additional guidance is available in our style and formatting guide Communications Psychology formatting guide.

We hope to receive your revised paper within 8-12 weeks; please let us know if you aren't able to submit it within this time so that we can discuss how best to proceed. If we don't hear from you, and the revision process takes significantly longer, we may close your file. In this event, we will still be happy to reconsider your paper at a later date, provided it still presents a significant contribution to the literature at that stage.

Best regards,

Marika

Marika Schiffer, PhD
Chief Editor
Communications Psychology

REVIEWER REPORTS:

Reviewer #1 (Remarks to the Author):

I liked many things about the initial submission but also felt that a number of issues were unclear, confusing, or not entirely convincing. These have now been clarified to my satisfaction. I believe that this revision is vastly improved, especially with the addition of Study 4, and I am therefore pleased to recommend it for publication. Congratulations to the authors on a nice piece of work.

Reviewer #2 (Remarks to the Author):

This paper is nicely written and presents an interesting theoretical perspective. Nevertheless, this cannot compensate for the methodological flaws and the disconnect between the theoretical claims and empirical data. The rebuttal letter ignores or misunderstands several important problems. Some of the most important points are briefly reiterated below. Please refer back to review comments in first round for more details.

Manipulation

The participants were asked to bear in mind a certain perspective and respond with this perspective in mind. Those who said they did not adopt the assigned perspective were excluded. This is quite similar to what is done in the false polarization literature, where people are asked how they think the other political side would respond to a set of items. No one would assume, for example, that how liberals think conservatives would respond is equivalent to the actual perspective of conservatism. On the contrary, the discrepancy between how conservatives respond and how liberals think they would respond (and vice versa) tends to be large. Similarly, it would be absurd to assume that how people imagine that a person holding a certain epistemic perspective (e.g., truth speaking) would respond is equivalent to what someone who genuinely endorses that perspective would respond. There is neither empirical evidence nor credible theoretical basis for the causal claims the authors make based on their manipulation.

Measure of epistemic beliefs

A previous review pointed out some problems with the usage of the Evidence-Intuition scale (e.g., it matches poorly with the theoretical constructs used in this paper). In their rebuttal letter, the authors ignored these problems and instead claim that this scale "is part of a supplementary exploratory analysis". However, a large portion of the results presented in the main text of the paper are based on this scale, so these problems certainly cannot be glossed over. What is more, these scales appear

to have an even more central role in the pre-registered hypotheses than the experimental manipulation does. For example, the first two hypotheses in all four pre-registrations are “Participants who prefer accuracy over sincerity are likely to view an accurate politician less favorably if the politician is breaking democratic norms” and “Participants who prefer sincerity over accuracy are likely to view sincere politicians favorably even if the politician is breaking democratic norms.” In fact, most of the pre-registered hypotheses are not about causal effects, contrary to what the authors claim.

Risk for Type I error

The authors misunderstand the problem with inflated type I error rates and misrepresent the paper by Cramer et al. (2015), which discusses the exploratory use of multiway analysis of variance. What they seem to do in this revised paper is to apply Benjamini-Hochberg corrections to all effects in their ANOVAs. In a confirmatory approach, controls should be applied specifically to all significance tests that bear on the research hypotheses across different analyses within each study (not all effects within each analysis of variance, as if the tests were exploratory). Cramer et al. (2015) do not make the strange claim that pre-registration alone would be a remedy to this problem. What they write is exactly what has been pointed out here: “By preregistering their studies and their analysis plans, researchers are forced to specify beforehand the exact hypotheses of interest. In doing so, as we have argued earlier, one engages in confirmatory hypothesis testing (i.e., the confirmatory multiway ANOVA), a procedure that can greatly mitigate the multiple-comparison problem. For instance, consider experimental data analyzed with a $2 \times 2 \times 3$ multiway ANOVA; if the researcher stipulates in advance that the interest lies in the three-way interaction and the main effect of the first factor, this reduces the number of tested hypotheses from seven to two, thereby diminishing the multiplicity concern...” Again, controls should be applied only to significance tests directly bearing on hypotheses in a confirmatory study.

Pre-registration

The authors claim that their studies were pre-registered. While pre-registrations were made for all of the studies, the hypotheses presented in the paper do not match the ones that were actually pre-registered. What is described as the “Perspective hypothesis” (H2-H7) in the paper only occurs in pre-registration for the fourth study, which was presumably conducted after the first round of reviews. This is the only hypothesis in any of the pre-registrations that clearly sets forth a causal hypothesis in line with the narrative of the paper. None of the three other pre-registrations contain any similar hypothesis – they are all correlational. The researchers have added an appendix describing which of the pre-registered hypotheses they claim test the hypotheses that are described in the paper, but this is in reality post hoc modification of hypotheses that undermines the claim that the hypotheses were pre-registered. While it can be acceptable to alter hypotheses or add new ones after pre-registration, this must be transparently and honestly acknowledged.

Conceptual precision

Claims that are made in the paper should be accurate and precise. For example, the authors were asked what evidence there is that people’s conceptions of truth and honesty have undergone significant changes in recent decades. The rebuttal letter contains a vague response (“The sentences in question serve to situate our work within the broader literature and establish the context for our study. They do not present new claims but rather summarize existing research to frame our contribution”) but no answer to the actual question has been provided in the rebuttal letter or the paper. It is one thing to say that research has shown that this is a problem today and another to say that research has demonstrated changes over time.

* **TRANSPARENT PEER REVIEW:** Communications Psychology uses a transparent peer review system. This means that we publish the editorial decision letters including Reviewers’ comments to the authors and the author rebuttal letters online as a supplementary peer review file. However, on author request, confidential information and data can be removed from the published reviewer reports and rebuttal letters prior to publication. If your manuscript has been previously reviewed at another journal, those Reviewers’ comments would not form part of the published peer review file.

** Visit Nature Research’s author and referees’ website at www.nature.com/authors for information about policies, services and author benefits**

Communications Psychology is committed to improving transparency in authorship. As part of our efforts in this direction, we are now requesting that all authors identified as ‘corresponding author’ create and link their Open Researcher and Contributor Identifier (ORCID) with their account on the Manuscript Tracking System prior to acceptance. ORCID helps the scientific community achieve unambiguous attribution of all scholarly contributions. You can create and link your ORCID from the home page of the Manuscript Tracking System by clicking on ‘Modify my Springer Nature account’ and following the instructions in the link below. Please also inform all co-authors that they can add their ORCIDs to their accounts and that they must do so prior to acceptance.

Version 3:

Decision Letter:

Dear Ms Huttunen,

Your manuscript titled "Subjective conceptions of honesty and tolerance for democratic norm violations" has now been seen by an additional reviewer, whose comments appear below. In light of their advice I am delighted to say that we are happy, in principle, to publish a suitably revised version in *Communications Psychology*.

We therefore invite you to revise your paper one last time to address the remaining concerns of our reviewers and a list of editorial requests. At the same time we ask that you edit your manuscript to comply with our format requirements and to maximise the accessibility and therefore the impact of your work.

In response to Reviewer #3's concerns, we ask that you clarify that your work only focuses on right-wing extremism/right-wing populism. We agree with the reviewers' observation that it is reasonable to make a distinction between the literature that gave rise to a question and the results' potential implications beyond that focus. You are not required to also provide a study on left-wing populism or left-wing extremism at this stage nor do we request a full exploration of mechanisms in either group. However, any unwarranted implication that left-wing extremism doesn't exist or would not potentially be characterized by similar processes needs to be avoided.

Editorially, we highlight that Results reporting in the main manuscript needs to be more comprehensive - at present many preregistered key analyses are only listed in the Supplement. These results Tables must be moved to the main text (and CIs added).

EDITORIAL REQUESTS:

SUBMISSION INFORMATION:

OPEN ACCESS:

*** TRANSPARENT PEER REVIEW:** *Communications Psychology* uses a transparent peer review system. On author request, confidential information and data can be removed from the published reviewer reports and rebuttal letters prior to publication. If you are concerned about the release of confidential data, please let us know specifically what information you would like to have removed. Please note that we cannot incorporate redactions for any other reasons.

*** CODE AVAILABILITY:** All *Communications Psychology* manuscripts must include a section titled "Code Availability" at the end of the methods section. We require that the custom analysis code supporting your conclusions is made available in a publicly accessible repository at this stage; please choose a repository that generates a digital object identifier (DOI) for the code; the link to the repository and the DOI must be included in the Code Availability statement. Publication as Supplementary Information will not suffice.

* DATA AVAILABILITY:

All *Communications Psychology* manuscripts must include a section titled "Data Availability" at the end of the Methods section. More information on this policy, is available in the Editorial Requests Table and at <http://www.nature.com/authors/policies/data/data-availability-statements-data-citations.pdf>.

Link Redacted

Best regards,

Marike

Marike Schiffer, PhD
Chief Editor
Communications Psychology

REVIEWERS' COMMENTS:

Reviewer #3 (Remarks to the Author):

In this timely and theoretically interesting manuscript, the researchers tested whether different perspectives on honesty – a focus on beliefs/authenticity or on facts – shaped people's responses to norm violations. The research tested and found support for the hypothesis that "when people are directed to rely on sincerity rather than accuracy as the main criterion of honesty, they became more tolerant of democratic norm violations."

This manuscript was characterized by a number of strengths. As noted above, the research is timely – it is of considerable interest and importance given shifting norms about societal discourse and the "backsliding" democratic norms. The work is also theoretically novel – it is grounded in literature on motivated reasoning and perspective taking, but moves that literature into a new direction. The research is methodologically sound, and the data analyses are rigorous, comprehensive, and well-reported. Taken together, the strengths of this research indicate it could make a meaningful contribution to existing literature.

Nonetheless, this manuscript was also marked by some meaningful limitations. Perhaps the most important limitation was a conceptual and interpretive one. The introduction and justification for the study asserts that the problem explored in this study – an emphasis on "belief-speaking" yielding acceptance for norm violation – is one that exists primarily on the right. In fact, the introduction paints a picture that this pattern exists *only* on the right. Although it is not useful to engage in "both-sideism" merely for the sake of doing so, I was struck by the notion that the left also engages in "belief-speaking" and that the introduction does not seem to recognize that. One anecdotal example: the oft-maligned notion that people should "speak their truths" is prominent among the left but not the right. Consistent with the notion that belief-speaking might exist on the left and right, research also indicates that liberals and conservatives similarly reject scientific evidence when it does not fit their worldview (see Washburn and Skitka), engage in confirmation bias, and engage in motivated reasoning. The literature review does not meaningfully explore the possibility that belief-speaking exists (perhaps in different forms) across the political spectrum, focuses narrowly on research documenting the rigidity of the right, and ignores research documenting motivated reasoning as a human rather than a politicized tendency. Still, an open-minded reader might be willing to accept the notion that there is ideological asymmetry in belief-speaking if the data bore that out. Alas, when one gets to the results of this studies, the data shows no such thing. The researchers do a deep-dive on these tendencies among participants on the right, but do no such analysis for participants on the left. Taken together, the data show, as the manuscript confesses, "This effect is present regardless of a person's political orientation or pre-existing epistemic preferences." Afterward, the discussion section of the manuscript returns to the notion that this is a problem of the right. Furthermore, given the centrality of political orientation in your introduction and discussion, it is notable that you only include the variable as a covariate in one of the four studies.

So, what could one do in response to this critique? In the introduction, you could either remove the focus on political orientation as a driver of this phenomenon, or you could broaden the focus to include the possibility that (as you find) this phenomenon exists along the entire political spectrum. If these changes feel inappropriate or misguided because they might lead "hypothesizing after the results are known", then the correction could exist within the discussion section. From my perspective, your fundamental finding IS interesting and important, even if you made the change to describe it as a tendency that exists among people in general and as one that is a risk to democratic norms.

The other limitation is a relatively minor one. There is a lack of clarity when it comes to the method. The manuscript reports four studies in one combined method and results section. Given the clear and meaningful similarities between the studies that seems like a reasonable choice. But the justification for each of the studies and their differences is difficult to ascertain – which makes it more difficult to make sense of the findings and would undermine attempts at replication. This could be cleared up with a paragraph and/or table that clarifies the distinctions between the studies and their purposes.

In sum, the core finding of the research was theoretically interesting, timely, and interesting. The research was well-designed (despite a lack of clarity), and the analysis was appropriate and comprehensive. The manuscript, nonetheless, would be

strengthened if it chose a more consistent, more accurate, and more theoretically grounded approach to motivated reasoning across the political spectrum rather than presenting it as a problem of the right.

replicably affects outcome Y (tolerance for norm violations), then that is illuminating and interpretable irrespective of whatever set of other individual-differences variables a, b, c, might also be associated with Y

3. **Methodological Concerns (Exclusion Criteria):** Reviewer 1 questioned the exclusion criteria related to participants' adoption of assigned perspectives. As we noted in the paper, the results do not change appreciably if we instead include all participants; we suspect this escaped the reviewer's notice. (We acknowledge that it would have been better to report the analysis with all participants in detail, but because we preregistered the conditional analysis, we saw no need at the time – we were wrong, but this is of course easily fixed. The ANOVA tables for all analyses with all participants can be found at <https://osf.io/4qt5w>).

We must add that this type of exclusion is standard in experimental research to ensure manipulation fidelity. It is known as a Per Protocol (PP) analysis, which is a widely accepted approach in experimental research. The reviewer, by contrast, advocates for an Intention to Treat (ITT) analysis where all participants are considered irrespective of whether they conformed to the protocol. Both methods have advantages and disadvantages, but PP analysis is appropriate when adherence to experimental protocols is critical to infer the underlying psychological processes (e.g., Tripepi, G., Chesnaye, N. C., Dekker, F. W., Zoccali, C., & Jager, K. J. (2020). Intention to treat and per protocol analysis in clinical trials. *Nephrology*, 25(7), 513–517. <https://doi.org/10.1111/nep.13709>). Because the PP vs ITT distinction is well known in the field, we saw no need to discuss our choices (and the underlying distinction) in the original submission. This omission is easily rectified by revision.

Response to Reviewer 2

1. **Experimental Manipulation:** The reviewer opines that there is no evidence that our intervention shifted people's perspective. Notably, this inverts Reviewer 1's criticism that our Results section "reads more like a series of manipulation checks".

In our view, we instantiated an intervention from a well-established toolbox of perspective-taking protocols (e.g., Vargiu, C., & Nai, A. (2022). Sincerity Over Accuracy: Epistemic Preferences and the Persuasiveness of Uncivil and Simple Rhetoric. *International Journal of Communication*, 16(0), 24. <https://ijoc.org/index.php/ijoc/article/view/18018>; Hahl, O., Kim, M., & Zuckerman Sivan, E. W. (2018). The Authentic Appeal of the Lying Demagogue: Proclaiming the Deeper Truth about Political Illegitimacy. *American Sociological Review*, 83(1), 1–33. <https://doi.org/10.1177/0003122417749632>). Moreover, we followed up by asking participants to self-report compliance, and we observed outcomes of the intervention that we hypothesized (and preregistered) based on prior theory.

We are at a loss to know what else the reviewer thinks we should have done. We are also at a loss to understand why the reviewer claims we present no evidence to back up our claims in the abstract, when in fact every assertion in the abstract is fully supported by our results.

2. **Hypotheses:** Our hypotheses were detailed in the preregistration (accessible to the reviewers), although we acknowledge that we did not explicitly cross-reference them (e.g., by number) with the Introduction and Results. We agree that we were not sufficiently explicit but this is an issue that can be readily addressed by revision.
3. **Statistical Issues (Type I and II errors and power):** The reviewer correctly notes that we conducted many tests within a 2 x 2 x 2 design and that therefore the issue of Type I errors must be considered. The reviewer also claims that "Without corrections [to significance levels], the results can be considered exploratory rather than confirmatory and further studies are needed to replicate them before any conclusions can be drawn."

The latter claim (a) ignores that we replicated the principal findings 3 times already and (b) is at odds with the relevant literature (e.g., Cramer, Angélique O. J., van Ravenzwaaij, D., Matzke, D., Steingroever, H., Wetzels, R., Grasman, Raoul P. P., Waldorp, Lourens J., & Wagenmakers, E.-J. (2015). Hidden multiplicity in exploratory multiway ANOVA: Prevalence and remedies. *Psychonomic Bulletin & Review*, 640–647. <https://doi.org/10.3758/s13423-015-0913-5>). Cramer et al. (2015) note that preregistration is an effective remedy to the multiple-comparison problem because it converts exploratory to confirmatory analyses. Given that we preregistered the hypotheses, inflation of the Type I error rate was largely already controlled in our original submission, contrary to the reviewer's concern.

Nonetheless, we have now applied two other remedies proposed by Cramer et al. to our data. First, to control for false discovery rate, we ran standard Bonferroni corrections. Second, we applied the Benjamini-Hochberg (BH) procedure (Benjamini, Y., & Hochberg, Y. (1995). Controlling the False Discovery Rate: A Practical and Powerful Approach to Multiple Testing. *Journal of the Royal Statistical Society Series B: Statistical Methodology*, 57(1), 289–300. <https://doi.org/10.1111/j.2517-6161.1995.tb02031.x>) and find that none of these significantly affect our conclusions. Specifically, the actual p-values change on both approaches, and the BH corrections make some of the main effects and interactions non-significant. Crucially, however, none of the effects of the perspective variable are affected by the corrections. (Our corrections were unduly conservative because we assumed throughout that we tested all hypotheses in the design, which ignores preregistration – this merely strengthens our point).

The multiple-comparison-adjusted ANOVA tables for all experiments are available at <https://osf.io/mw7cn>.

Concerning power analyses, the reviewer correctly notes their absence, and we apologize for that oversight. We have now conducted those analyses through power simulations in R (see <https://julianquandt.com/post/power-analysis-by-data-simulation-in-r-part-i/>) and find that the minimum sample size to detect small effects in our design is 25 participants per condition, which is less than we have in our experimental conditions.

It follows that both the Type I inflation and the lacking power analyses are not fatal problems but are easily resolved by revision.

4. **Problems with the measures:** Reviewer 2 expresses dissatisfaction with the Epistemic Evidence-Intuition Scale (E2IS; it used to be called EIS but we have changed the name). These measures were exploratory and not central to our primary hypotheses, as explicitly stated in the manuscript. None of our conclusions would change if we were to omit the E2IS. Moreover, the E2IS scale has been validated in a separate paper that is currently under review, and which was cited (as a preprint) in our original submission. We also note that some of the items have previously been used in explorations of honesty; see, e.g., Hahl, Oliver & Minjae, Kim & Sivan, Ezra. (2018). The Authentic Appeal of the Lying Demagogue: Proclaiming the Deeper Truth about Political Illegitimacy. *American Sociological Review*. 83. 000312241774963. 10.1177/0003122417749632.

The reviewer also expressed concern about the model misfit concerning our “perceived honesty” and “likeability” scores. It is correct that the confirmatory model in Experiments 2 and 3 did not fit terribly well, even though several fit indices (CFI, TFI, SRMR) were within an acceptable range. However, it is difficult to see how this problem would be fatal: if we instead form composite scores that average across all items, which is common practice in the field, our main results are unchanged.

5. **Lack of Transparency:** Reviewer 2 expressed concerns about transparency, particularly regarding data/code availability. In response, we must first point out that the initial submission provided the AsPredicted IDs for all preregistrations on p. 15.

We also provided a link to raw data and analysis scripts (<https://osf.io/tkr56/>) in the Nature Reporting Summary, as requested. However, the paper was sent out for review before we had returned the requested form within the deadline provided. (Specifically, the form was requested on 9 October and we were given 2 business days to respond. We returned the form on 11 October, and we received an email on 14 October confirming the receipt of these, with the note that “Unfortunately since your manuscript is already with reviewers we are unable to update the article file at this stage, but please don’t worry as you will be able to do so yourself at a later stage and the statements are not crucial to include for the peer review process”.)

We apologize that the appropriate form was not submitted together with the paper, but equally, it appears that Reviewer 2’s point about lacking transparency may have arisen out of a clerical glitch over which we had no control.

Either way, our paper meets all applicable transparency requirements and all data, analysis scripts, and preregistrations are publicly available as detailed in the Reporting Summary.

We remain confident in the contribution of our manuscript and believe that the reviewers’ comments reflect an opportunity to clarify and strengthen our work. However, we respectfully request reconsideration of your decision to reject the manuscript. As we have shown in this letter, we have clear answers or rebuttals to all of the reviewers’ criticisms.

We greatly appreciate your time and effort and look forward to your response.

Best wishes, Steve

Professor Stephan Lewandowsky FAcSS FAPS

School of Psychological Science

University of Bristol

12A Priory Road

Bristol BS8 1TU

Bluesky: @lewan.bsky.social

Mobile/WhatsApp: +44 74401 89544

Access to >80 global experts in misinformation research: <https://sks.to/misinfoexperts>

TeDCog research group: <https://sks.to/tedcog>

Homepage: <https://www.lewan.uk>

Top 10 UK university and 62nd in the world (QS Rankings 2022)

Top 5 UK university for research (THE analysis of REF 2014)

Top 10 most targeted universities by top UK employers (High Fliers 2021)

RESPONSE TO REVIEWER COMMENTS:

Reviewer #1 (Remarks to the Author):

Review of "The evolution of truth in political discourse from fact to feeling..."

I have read this paper with considerable interest and found much to like about it. At the same time, I think that much more needs to be done to illuminate the role of ideological and psychological variables (see below).

The introduction provides a very useful summary and review of social scientific research on how right-wing populists around the world have contributed to an informational climate in which objective considerations such as fact and evidence are devalued and subjective feelings about what is true are prioritized. I was surprised, however, given that this manuscript has been submitted to a psychology journal, that there was really no discussion of theory and research in social, personality, or political psychology. Consequently, no satisfying psychological account of why right-wing populists focus on feelings rather than facts was provided.

Fortunately, this problem is easily remediable, because a lot of research in political psychology has documented left-right asymmetries in epistemic motives and abilities. In Chapter 7 of a book entitled *Left & Right: The Psychological Significance of a Political Distinction*, Jost (2021, pp. 230-281) reviewed dozens of studies documenting associations between ideology and various measures of epistemic preferences that are very close to the Evidence-Intuition Scale used in the present research.

Response: We thank the reviewer for the point about including background on political left-right asymmetries in epistemic motives. While the core arguments of our manuscript do not center on left-right asymmetries, we recognize that this literature is relevant to the broader discussion. As a response to your feedback, we have added a section in the introduction that better situates our work within research on epistemic differences across the ideological spectrum. We trust this addition strengthens the manuscript by clarifying its theoretical positioning while maintaining the original scope of our argument. We have also added a further experiment, explained below, that considers political orientation explicitly.

Subjective conceptions of honesty and tolerance for democratic norm violations

This may be something dictated by the journal's formatting instructions, but I did not like the fact that the descriptions of research methods, participants, materials, and procedure came at the very end of the paper. I did not find it possible to understand or interpret the results section without finding the methods section first. The transition from the introduction to the results was abrupt, to say the least.

Response: Thank you for your feedback on the manuscript structure. The placement of the methods section at the end follows the journal's formatting guidelines. To improve clarity and ease of interpretation, we have revised the introduction-to-results transition while still conforming to the journal's guidelines. We hope this makes the manuscript more accessible while adhering to the required format.

In any case, looking carefully at the methods section, I found that I did not like the exclusion criteria that were apparently pre-registered. The researchers appear to have excluded any participants who stated that they did not adopt the assigned perspective. This is not a common way of dealing with experimental manipulations, which can surely affect people without their full understanding of how or why. What if some participants were affected by the manipulation but thought (or said) they weren't? Is it worth destroying the power and beauty of random assignment to drop participants based on their subjective reports about what they thought they were doing, given that they can be wrong what they did? I don't think so.

As it stands, the researchers dropped 15% of the participants from Study 1, 19.5% from Study 2, and 6.5% from Study 3. I would much rather see analyses based on the full samples in the main text (with these analyses based on smaller samples included in an online supplement as robustness checks), because only then is random assignment preserved and there is greater statistical power to test hypotheses. I fear that pre-registering and sticking rigidly to exclusion criteria based on hunches is doing much more harm than good in experimental social psychology.

Response: We appreciate the feedback. However, we would like to clarify our analytic choices. Our main analysis is known as a Per Protocol (PP) analysis, which is a widely accepted approach in experimental research. What seems to be requested by the reviewer is an Intention to Treat (ITT) analysis where all participants are considered irrespective of whether they conformed to the protocol. Both methods have advantages and disadvantages, but PP analysis is appropriate when adherence to experimental protocols is critical to infer the underlying psychological processes (e.g., Tripepi, G., Chesnaye, N. C., Dekker, F. W., Zoccali, C., & Jager, K. J. (2020). Intention to treat and

per protocol analysis in clinical trials. *Nephrology*, 25(7), 513–517. <https://doi.org/10.1111/nep.13709>). We have also clarified this in the manuscript.

As we had pre-registered the analyses and hypotheses under the PP analysis, we initially saw no need to report the full ANOVA results for the ITT analysis, in particular because the ITT analysis does not significantly change the results. However, in light of the reviewer's comments, we have revised the manuscript to clarify our position and the distinction between PP and ITT analyses, and we now also include the unfiltered (ITT) analyses in the online supplement. We hope that making both analyses available resolves the reviewer's concerns.

I cannot tell from either the meager methods section or the online supplement if the researchers measured left-right political orientation (or ideology), but I hope so. There are several reasons for this, including the fact that the entire introduction is about right-wing populism (and not left-wing populism, which is mentioned only in a footnote), and the fact that there is lots of evidence showing that leftists and rightists differ in epistemic motives and abilities (Jost, 2021).

To me, the results section does not really illuminate anything unless the authors consider the role of ideology. Based on previous research, one would surely expect that conservative-rightists would be more likely to value intuition and less likely to value evidence, compared to liberal-leftists. This would presumably make conservative-rightists more tolerant of dishonesty, more likely to favor "belief-speaking," more accepting of power expansion and norm violations, etc. Demonstrating all of this would be very interesting and useful for understanding political psychology.

Response: Although we agree with the comment about left-right asymmetries, these were not at the core of our work. The focus was –and continues to be– on ontologies of truth and epistemic preferences. In response to the reviewer's concern, we have conducted a further experiment where political orientation was measured which is reported in the revision. In a nutshell, while we find political leaning to have a significant main effect, as expected on the basis of the literature cited by the reviewer, the effects of the perspective intervention are neither eliminated nor moderated by political leaning. . Political orientation appears to play a similar role to our measured epistemic preferences. We believe that these new results have strengthened the manuscript and have further embedded our work into the existing literature.

As it stands, the results section is not very enlightening and reads more like a series of manipulation checks (e.g., people who were told that Mr. Smith was lying and violating norms perceived him as less honest than people who were told he was not lying or violating norms). than theoretically meaningful results. I would also like to see much more use made of individual (and ideological) differences in EIS, which was mentioned only in passing as a “covariate” on p. 11.

I think that this research program has potential, but only if it can be used to illuminate ideological asymmetries and psychological processes involved in democratic backsliding. Otherwise, it is not clear to me what is being explained here. The fact that 80% of research participants read the study materials carefully enough to draw appropriate conclusions about hypothetical candidates is not, in and of itself, much of a scientific discovery, it seems to me.

Results: We appreciate the reviewer’s perspective on the potential future avenues for this research. However, our primary focus is not on ideological asymmetries but rather on ontologies of truth—how individuals conceptualize and evaluate truth claims in political contexts. While epistemic preferences are included as a secondary analysis, the core contribution of our study lies in examining how interventions that change the conception of honesty people adopt shape perceptions of political norm violations, rather than in mapping ideological differences per se.

To enhance the clarity and theoretical contribution of our results, we have revised the manuscript to better align with our overarching research aims. We have also refined our discussion to more explicitly articulate what our findings reveal about the cognitive mechanisms underlying political truth perceptions and their implications for democratic norms. We hope these revisions strengthen the manuscript and clarify the unique contributions of our approach.

Reviewer #2 (Remarks to the Author):

This is a very well-written paper on a timely and important topic. The distinction between fact-speaking and belief-speaking is likely to be important. Nevertheless, the reported studies suffer from methodological problems, which make the paper unsuitable for publication in this journal. The most important problems are listed below:

Subjective conceptions of honesty and tolerance for democratic norm violations

1. The manipulation

The authors conclude that whether a person takes a “belief-speaking” or “fact-speaking” perspective affects acquiescence to democratic norm violations and politicians’ dishonesty. They presented participants with a vignette describing the perspective of “Sam” (for belief-speaking) or “Taylor” (fact-speaking) and asked the participants to adopt this perspective when answering subsequent questions (“Please take the perspective of Sam[Taylor] when answering the question that will be shown on the next page”... “Bearing in mind your perspective of Sam/Taylor, please answer the following questions.”). According to the pre-registrations “Participants who indicate that they have not adopted their assigned perspective will also be excluded from analyses.” Nevertheless, it is not obvious how the request to adopt another person’s perspective was interpreted. It is quite possible that the participants were simply trying to imagine how a person with a belief-speaking or fact-speaking perspective would react to democratic norm violations and dishonesty—or in other words, that the results reflect the participants’ implicit theories about people with these perspectives, rather than their own reactions to norm violations and dishonesty from a different epistemological perspective. The paper does not present any evidence that the actual perspectives of the participants were shifted by the manipulation.

Response: Participants were presented with a vignette describing the perspective they were asked to take, as reported in the Results -section. The vignettes are attached at the end of the paper and the procedure is described in the Methods -section. We now additionally clarify how a participants’ adoption (or not) of the intended perspective was confirmed. We have further reviewed the manuscript to ensure that the reporting is clear. We hope that our revision addresses the reviewer’s concerns.

Some of the conclusions go even further: “Our data shows for the first time that this preference for belief-speaking can be experimentally manipulated and has drastic flow-on consequences for the acceptance of democratic norm violations. This finding supports the notion that belief-speaking is a fundamental aspect of right-wing populist communication.” No evidence is presented to back up any of these strong claims.

Response: Thank you for your feedback on our conclusions. We would like to clarify that the evidence supporting these claims is presented in the main text (particularly the “Results” and “Discussion” -sections) and further supported with the detailed evidence presented in the online supplement. Specifically, our experimental results demonstrate the impact perspective-taking has on participants’ acceptance of norm violations.

Nonetheless, we have revisited the relevant sections and adjusted wording where necessary to better reflect the strength of our findings. We hope these revisions address the reviewer's concerns and link our claims more clearly to the data.

2. The hypotheses

The hypotheses are not made explicit in the paper. The authors write: "We present a series of preregistered studies that examined the conditions under which people acquiesce to democratic norm violations and politicians' dishonesty. We find that when participants are asked to take a perspective of honesty that emphasises sincerity over accuracy, which we call 'belief-speaking', they are more willing to accept norm violations by politicians than if participants take a perspective that emphasizes accuracy as a criterion for honesty, which we call 'fact-speaking'." They thereafter present a large number of significance tests of effects based on a complex design without making explicit how these correspond to their hypotheses. Furthermore, the hypotheses that are listed in the pre-registrations are different from what one might guess that they would be based on the paper—they appear to mainly concern effects of individual differences (e.g., "Participants who prefer accuracy over sincerity are more likely to...")—and some of the significance tests that are reported prior to the exploratory section are completely unrelated to these hypotheses. It is hard to figure out which statistical tests were hypothesis tests and which ones were not.

Response: We acknowledge the oversight in reporting of hypotheses in the original manuscript. While the hypotheses were available in the pre-registration files, the manuscript did not make the necessary links terribly clear. The manuscript has been revised to ensure a clear link between the hypotheses, analyses, and the conclusions drawn from the data. We believe that this strengthens the manuscript.

3. Type I and Type II errors

The authors report a large number of significance tests based on a 2*2*2 between-subjects design without specification of corrections for the number of statistical hypotheses to prevent an inflated risk of Type I errors. We must assume that there were no corrections since the authors verified that they have described all corrections when submitting the paper. Without corrections, the results can be considered exploratory rather than confirmatory, and further studies are needed to replicate them before any conclusions can be drawn. It is obvious that if appropriate corrections were made, many of the results, including almost all interactions, would not be non-significant (many of the p-values are marginally below .05).

Response: We agree that the issue of Type I errors must be considered due to the design. However, in regards to the comment of “[w]ithout corrections [to significance levels], the results can be considered exploratory rather than confirmatory and further studies are needed to replicate them before any conclusions can be drawn.” we respectfully disagree on two principal grounds;

- 1) The principal findings are replicated four times across our four experiments.
- 2) The comment is at odds with the relevant literature (e.g., Cramer, Angélique O. J., van Ravenzwaaij, D., Matzke, D., Steingroever, H., Wetzels, R., Grasman, Raoul P. P. P., Waldorp, Lourens J., & Wagenmakers, E.-J. (2015). Hidden multiplicity in exploratory multiway ANOVA: Prevalence and remedies. *Psychonomic Bulletin & Review*, 640–647. <https://doi.org/10.3758/s13423-015-0913-5>). In particular, Cramer et al. (2015) note that preregistration is an effective remedy to the multiple-comparison problem because it converts exploratory to confirmatory analyses. Given that we preregistered the hypotheses, inflation of the Type I error rate was largely controlled in our original submission, contrary to the reviewer’s concern.

Nonetheless, we have now applied two additional remedies proposed by Cramer et al. (2015). First, to control for false discovery rate, we ran standard Bonferroni corrections. Second, we applied the Benjamini-Hochberg (BH) procedure (Benjamini, Y., & Hochberg, Y. (1995). Controlling the False Discovery Rate: A Practical and Powerful Approach to Multiple Testing. *Journal of the Royal Statistical Society Series B: Statistical Methodology*, 57(1), 289–300. <https://doi.org/10.1111/j.2517-6161.1995.tb02031.x>) and find that none of these significantly affect our conclusions. While the actual p -values change on both approaches and the BH corrections make some of the main effects and interactions non-significant (all results are presented in the online supplement as part of the full ANOVA tables), none of the effects of the crucial perspective variable are affected by the corrections. To strengthen our point, our corrections were unduly conservative because we assumed throughout that we tested all hypotheses in the design, which ignores preregistration.

The sample sizes are small for a complex design with tests of between-group interactions, leading to an inflated risk of Type II error. No relevant power analysis is reported. It seems that statistical power was not considered. In one of the pre-registrations, the authors write: “We will recruit 300 participants as correlations generally stabilize around 250 and there are eight experimental conditions”. Although this is a relevant consideration in research on individual differences, it is separate from the power issue, and it is not directly applicable to the estimation of parameters in a complex experimental design.

Response: We appreciate the concern and apologize for the oversight. However, we are pleased to confirm that as per power analysis simulations over 10,000 iterations for 0.85 power, the minimum required sample size is 25 participants per condition, which is less than we had in our experimental conditions for Experiments 1-3. To further address the issue, prior to conducting the additional experiment (Experiment 4) we ran a power analysis simulation. It showed that to detect a smallest effect size of interest (SESOI) of partial eta-squared 0.04 with 90% power would require 50 participants per condition, which we achieved. Thus, while we acknowledge the absence of power analysis reporting in the original manuscript and pre-registrations, we trust that the matter has been resolved in the revision process.

4. Measures

The authors use the Evidence-Intuition scale to operationalize pre-existing views of honesty. Leaving aside weaknesses of this scale aside, there are two problems. 1. The authors appear to collapse the three scales without providing any evidence that these have a substantive general factor. 2. The link between the theoretical conception of preference for accuracy over sincerity and the content of the scales is somewhat tenuous. The scales concern faith in intuitive hunches about facts, belief that evidence and facts are important when telling whether something is true, and belief that truth is determined by those with political power. Although these are related to a preference for accuracy, what they have to do with beliefs about honesty and sincerity is less clear.

Response: We appreciate the comment but would like to elaborate on the role of the Epistemic-Evidence Intuition scale in the paper; it is part of a supplementary, exploratory analysis. The construction of the scale is not a part of this paper but the scale is used as it contains items examining people's preferences for accuracy and sincerity. Nonetheless, all analyses relating to the scale are exploratory and reported as such. These items have previously been used in e.g., Hahl, Oliver & Minjae, Kim & Sivan, Ezra. (2018). The Authentic Appeal of the Lying Demagogue: Proclaiming the Deeper Truth about Political Illegitimacy. *American Sociological Review*. 83. 000312241774963. 10.1177/0003122417749632.

The "perceived honesty" score (based on factor analyses) of judgments of Mr Smith covers both attributes such as authenticity, sincerity, and genuineness and attributes such as competence and accuracy. This labeling is somewhat confusing given the theoretical distinction between sincerity and accuracy. Furthermore, what is the reason for the model misfit (reported in a supplement)?

Response: We thank you for the feedback and have adapted the manuscript to more clearly reflect the interconnectedness of sincerity and accuracy as components of truth. We also acknowledge that the confirmatory model in Experiments 2, 3, and 4 does not fit terribly well, even though several fit indices (CFI, TLI, SRMR) were within an acceptable range. However, even if we instead form composite scores that average across all items, which is common practice in the field, our main results for all experiments are unchanged. Thus, we argue that the model misfit is not a cause for concern because our results are robust to using the alternative aggregating approach instead.

5. Transparency

There are no valid links to pre-registrations, no code availability statement, no clear description of hypotheses or corrections, and no description of the exact sample size for each experimental condition in the paper.

Response: We must point out that the initial submission provided the AsPredicted IDs for all pre-registrations (which include hypotheses and analyses plans) on p. 15. Those IDs are sufficient to access the pre-registrations from the AsPredicted website. Further to this, the code availability statement was part of the Nature Reporting Summary, which was provided alongside the manuscript at the time of submission. All the materials (including raw data and codes) were available at <https://osf.io/tkr56/>, as noted in the Nature Reporting Summary. In the revision, all links are now directly available in the manuscript. We believe that these concerns should therefore have been addressed

6. Other claims

What evidence is there for this claim: "People's subjective conceptions of truth and honesty have undergone significant changes in recent decades. Parts of society increasingly favour the sincere expression of personal belief, however inaccurate, over verifiable facts."?

Response: The sentences in question serve to situate our work within the broader literature and establish the context for our study. They do not present new claims but rather summarize existing research to frame our contribution. The relevant citations and supporting evidence for these points are provided throughout the introduction. We have reviewed the manuscript to flesh this out further.

(section 9 of the online supplementary material) that highlight the applicability of the E2IS scale. We must point out, however, that the E2IS is not central to our experiments and we are surprised it has attracted so much attention – our paper is reporting 4 experiments whose outcomes are informed by, but not critically determined by the E2IS, as we now explicate in detail below.

Your hypotheses were preregistered and you applied corrections for multiple comparisons. In addition, please ensure that your reporting and interpretation of the results, and especially of non-significant findings, follow the journal's standards, as summarized here: <https://www.nature.com/commspsychol/submit/submission-guidelines#statistical-guidelines> and referenced in greater detail in the attached Editorial Requests Table.

Response: We have followed the journal's standards in reporting and interpretation of statistical findings.

Round 2 reviews

Reviewer #1 (Remarks to the Author):

I liked many things about the initial submission but also felt that a number of issues were unclear, confusing, or not entirely convincing. These have now been clarified to my satisfaction. I believe that this revision is vastly improved, especially with the addition of Study 4, and I am therefore pleased to recommend it for publication. Congratulations to the authors on a nice piece of work.

Response: We thank the Reviewer for their valuable feedback and positive assessment of our work.

Reviewer #2 (Remarks to the Author):

This paper is nicely written and presents an interesting theoretical perspective. Nevertheless, this cannot compensate for the methodological flaws and the disconnect between the theoretical claims and empirical data. The rebuttal letter ignores or misunderstands several important problems. Some of the most important points are briefly reiterated below. Please refer back to review comments in first round for more details.

Response:

We appreciate the reviewer's engagement and the opportunity to clarify our work. However, we respectfully disagree with the suggestion that our rebuttal ignored or misunderstood the concerns raised. We have responded to each point in detail, both in the initial and current rounds, and we have endeavoured to understand the reviewers' concerns.

We have revised the manuscript to clarify the role of the experimental manipulation, the use of the scale, and the alignment between our theoretical claims and empirical results. We have also taken care to ensure that our conclusions and methodological descriptions are clear and transparent throughout.

Nonetheless, some points of disagreement may remain as we were unable to align some of the reviewer's comments either with the contents of the paper or scientific practice; we highlight those below.

Manipulation

The participants were asked to bear in mind a certain perspective and respond with this perspective in mind. Those who said they did not adopt the assigned perspective were excluded. This is quite similar to what is done in the false polarization literature, where people are asked how they think the other political side would respond to a set of items. No one would assume, for example, that how liberals think conservatives would respond is equivalent to the actual perspective of conservatism. On the contrary, the discrepancy between how conservatives respond and how liberals think they would respond (and vice versa) tends to be large. Similarly, it would be absurd to assume that how people imagine that a person holding a certain epistemic perspective (e.g., truth speaking) would respond is equivalent to what someone who genuinely endorses that perspective would respond. There is neither empirical evidence nor credible theoretical basis for the causal claims the authors make based on their manipulation.

Response:

We appreciate the reviewers' concerns and welcome the opportunity to further clarify how our manipulation functions in the paper.

To recap, we ask participants to adopt a particular perspective on honesty - not to speculate how another person would respond, but to temporarily reason from that perspective themselves, when evaluating the politician's behaviour. We excluded participants who indicated that they did not adopt the assigned perspective, as stated in the preregistration and clarified in the Methods section. This ensures that analyses reflect only those who complied with the manipulation. (However, we also report the full data in an additional analysis to confirm that our exclusion does not qualitatively alter the results – it does not, as we show in the OSM). As we explained in detail at the previous round, both analyses are well known and common in the literature. Our main analysis is known as a Per Protocol (PP) analysis,

and the analysis with all participants is an Intention to Treat (ITT) analysis; the former is particularly appropriate when adherence to experimental protocols is critical to infer the underlying psychological processes.

We consider our intervention to be another instance of common interventions that use perspective-taking-manipulations to examine how different lenses influence moral or political evaluations. Examples of these include but are not limited to:

Adida, C. L., Lo, A., & Platas, M. R. (2018). Perspective taking can promote short-term inclusionary behavior toward Syrian refugees. *Proceedings of the National Academy of Sciences of the United States of America*, *115*(38), 9521–9526. [10.1073/pnas.1804002115](https://doi.org/10.1073/pnas.1804002115).

Gennaro, G., Derksen, L., Abdelrahman, A., Broggin, E., Green, M. A., Haerter, V. A., Heer, E., Heidler, I., Kauer, F., Kim, H. N., Landry, B., Levis, A., Li, J., Şimşir, Ş., Srbinovska, I., Vital, R. A., Donnay, K., Gilardi, F., & Hangartner, D. (2025). Counterspeech encouraging users to adopt the perspective of minority groups reduces hate speech and its amplification on social media. *Scientific Reports*, *15*(1), 1–8. <https://doi.org/10.1038/s41598-025-05041-w>

Bruneau, E. G., & Saxe, R. (2012). The power of being heard: The benefits of ‘perspective-giving’ in the context of intergroup conflict. *Journal of Experimental Social Psychology*, *48*(4), 855–866. <https://doi.org/10.1016/j.jesp.2012.02.017>

We agree with the reviewer’s statement that “No one would assume, for example, that how liberals think conservatives would respond is equivalent to the actual perspective of conservatism”. However, we are unsure why that is relevant as the statement does not describe our intervention and is therefore an imperfect analogy. At no point do we ask a group of people to presume anything about what another group might do: instead, our intervention asks participants to emphasize one or the other component of honesty, both of which people are demonstrably familiar with.

We now demonstrate this familiarity in the paper in a new paragraph in the *Potential limitations* section:

“It is unlikely, and we do not claim, that our intervention altered participants’ fundamentally held beliefs about honesty. This is also unnecessary: our assumption was that people generally are conversant with both aspects of honesty and are therefore readily able to prefer one or the other when directed to do so by the intervention. There is support for that assumption both within our data and in the existing literature. First, the vignettes used here have been successfully used in previous research (asking people to pick which one they prefer), which showed that people are conversant with the two components of honesty because their preferences for one or the other vignette correlated with the same indicators of epistemic preferences that we used in Experiment 1 here (Vargiu & Nai, 2022). In further support, it has been shown that people adjust to fact-speaking or belief-speaking rhetoric simply by being asked to respond to text that is predominantly accuracy- or sincerity-based (Carrella et al., 2025). Similarly, when asked to judge which component of honesty is prevalent in a given text, people’s judgments are closely

aligned ($AUC \approx .80$) with the results of a computational linguistic analysis using prevalidated dictionaries (Carrella et al., 2025; Lasser et al., 2023). People can also readily generate times from their lives when they were authentic or sincere but did not tell the truth (Bailey & Iyengar, 2022), further attesting to people's familiarity with those two aspects of honesty. Moreover, we reported at the outset how identifiable lies can become acceptable when cloaked as authenticity (Hahl et al., 2018), precisely as expected if people are conversant with those two forms of honesty. Overall, there is little doubt that people are familiar with the distinction between sincerity and factual accuracy and can demonstrably recognize or adapt one or the other perspective on honesty in a variety of tasks and in response to a variety of cues." (Argument continues, but this paragraph captures the most important point and shows that our intervention is nothing like asking liberals to infer what conservatives might think).

To summarize, the reviewer's suggestion that our findings merely reflect participants' theories about others, rather than their own perspective, is inconsistent with both our instructions and evidence from the literature that people can recognize and adapt these perspectives.

Measure of epistemic beliefs

A previous review pointed out some problems with the usage of the Evidence-Intuition scale (e.g., it matches poorly with the theoretical constructs used in this paper). In their rebuttal letter, the authors ignored these problems and instead claim that this scale "is part of a supplementary exploratory analysis". However, a large portion of the results presented in the main text of the paper are based on this scale, so these problems certainly cannot be glossed over. What is more, these scales appear to have an even more central role in the pre-registered hypotheses than the experimental manipulation does. For example, the first two hypotheses in all four pre-registrations are "Participants who prefer accuracy over sincerity are likely to view an accurate politician less favorably if the politician is breaking democratic norms" and "Participants who prefer sincerity over accuracy are likely to view sincere politicians favorably even if the politician is breaking democratic norms." In fact, most of the pre-registered hypotheses are not about causal effects, contrary to what the authors claim.

Response:

We thank the reviewer for their continued engagement with our manuscript and their interest in the role of the Epistemic Evidence-Intuition scale (E2IS). We welcome the opportunity to further clarify the role of E2IS in our work. The scale was developed and validated in a separate paper (Abels, C., & Lewandowsky, S. (2024). Development and Validation of the Evidence Intuition Scale. <https://doi.org/10.31234/osf.io/u4xka>), and we consider its construction and psychometric evaluation to be beyond the scope of the present paper. However, we now briefly summarize the scale further in the Supplement, specifically section 9, including a summary report of the data supporting the scale.

We cannot, however, accept the reviewer's claims about our hypotheses and causality.

We must reiterate that the E2IS is not used to test any of our primary hypotheses, nor is it essential to our operationalization of our core intervention. Our main claims are based entirely on the effects of experimentally manipulated variables: the framing of perspective (fact-speaking vs. belief-speaking) and the politician's behaviour. While there is conceptual overlap in what the E2IS can be used to measure and what our work focuses on, our experimental manipulation of fact-speaking and belief-speaking are independent of the use of the scale.

Contrary to the reviewer's claim, our hypotheses are *all* about causal effects of experimental variables: the presence or absence of causality is not a function of which particular verb is being used in a preregistration ("prefer" may not be the reviewer's ideal choice, but it captures the result of our intervention – which is that participants are expected to then *prefer* the assigned perspective for the duration of the experiment), but is a function of the experimental design and random assignment of participants to conditions that only differ with respect to the intervention (we take up the exception, Experiment 1, below). We are not aware of any literature or precedent suggesting that the causality of an effect is determined by its linguistic description rather than the experimental design.

It might be helpful in this context to entertain a brief thought experiment: none of our conclusions would be altered in any way if we produced a hypothetical paper without Experiment 1 (the only one in which a subscale of the E2IS was used to inform subject assignment to condition) and without any mention of the E2IS and the exploratory analyses involving the E2IS in any of the remaining studies. This hypothetical paper, without any mention of the E2IS whatsoever, would still report 3 replications of our causal effect of the perspective-taking manipulation on people's endorsement of norm violations, and it would come to exactly the same conclusions. We are not suggesting that this hypothetical paper would be preferable, but it is important to recognize the subordinate role of the E2IS vis-a-vis our conclusions.

Specifically, we used the E2IS in two distinct and limited ways. First, in Experiment 1 we utilize a subset of items from the scale - specifically focused on inherent preferences for accuracy versus sincerity - to allocate participants' to experimental condition. These items come from previous published work by Vargiu & Nai (2022) and are conceptually aligned with the perspective manipulation that follows. We used the items to amplify our experimental manipulation in our first experiment. This choice was made because in the absence of any relevant precedent, we were interested in using the strongest-possible manipulation (i.e., confounding prior attitude with an amplifying intervention) to observe the consequences. When we found a strong effect in Experiment 1, we dropped the E2IS items from the intervention, and the remaining studies (i.e., 3 out of 4 studies) *did not use the E2IS anywhere except in exploratory ANCOVA analyses* to examine whether participants' inherent epistemic preferences influence their responses. These analyses are clearly labeled as exploratory and our central conclusions do not rely on those analyses – although of course they add helpful context.

We have revised the manuscript to clarify the distinction between the experimental manipulation and the E2IS, to specify the origin and prior use of parts of the scale and to ensure that its supplementary exploratory role is clear throughout.

Risk for Type I error

The authors misunderstand the problem with inflated type I error rates and misrepresent the paper by Cramer et al. (2015), which discusses the exploratory use of multiway analysis of variance. What they seem to do in this revised paper is to apply Benjamini-Hochberg corrections to all effects in their ANOVAs. In a confirmatory approach, controls should be applied specifically to all significance tests that bear on the research hypotheses across different analyses within each study (not all effects within each analysis of variance, as if the tests were exploratory). Cramer et al. (2015) do not make the strange claim that pre-registration alone would be a remedy to this problem. What they write is exactly what has been pointed out here: "By preregistering their studies and their analysis plans, researchers are forced to specify beforehand the exact hypotheses of interest. In doing so, as we have argued earlier, one engages in confirmatory hypothesis testing (i.e., the confirmatory multiway ANOVA), a procedure that can greatly mitigate the multiple-comparison problem. For instance, consider experimental data analyzed with a $2 \times 2 \times 3$ multiway ANOVA; if the researcher stipulates in advance that the interest lies in the three-way interaction and the main effect of the first factor, this reduces the number of tested hypotheses from seven to two, thereby diminishing the multiplicity concern..." Again, controls should be applied only to significance tests directly bearing on hypotheses in a confirmatory study.

Response:

We thank the reviewer for their careful reading and clarification.

We respectfully disagree that we misrepresented the Cramer et al. paper.

We said this in the paper: "Although most of our statistical tests involved preregistered hypotheses, which guards against the inflation of Type I errors associated with multiple statistical tests (Cramer et al., 2015), ..." We consider this to be an accurate (though highly condensed) summary of the extended quote from Cramer et al. provided by the reviewer above. We are unable to see how our phrase "... guards against the inflation of Type I errors associated with multiple statistical tests" is misrepresenting Cramer et al.'s "... can greatly mitigate the multiple-comparison problem".

To further guard against possible misinterpretation, we discussed this case with Prof E. J. Wagenmakers (the last author on Cramer et al. 2015) at length (correspondence available to the Editor upon request with Prof Wagenmaker's permission) and our response has been shaped by his comments.

Several points must be made with respect to this issue:

- Preregistered tests are conceptually different from other, exploratory tests. Any corrections should therefore be applied separately for confirmatory vs. exploratory tests.
- However, in a multi-experiment situation, an argument can be made that significant effects from earlier studies, even if originally exploratory, are no longer truly exploratory in subsequent studies (Wagenmakers, personal communication). At least tacitly, any existing effect should expect to be replicated, which means it is implicitly hypothesized and hence becomes confirmatory in subsequent experiments (to say otherwise would imply that we do not expect effects to replicate). This complicates the choice of adjustments.
- The litmus test of any effect is ultimately its replicability.
- The litmus test of any analysis is its robustness to assumptions such as adjustments for multiplicity.
- The analysis has to remain accessible and elegant.

To satisfy these constraints, we now report the analyses in the main body with the Benjamini-Hochberg corrections applied to *all* effects within each ANOVA model, treating the design conservatively and as if it were entirely exploratory in all experiments. We additionally assure the reader that if we apply a Bonferroni correction only to the confirmatory tests, we obtain the same conclusions.

We recognize that this approach is one of several possible options, but in our view it has the advantage that it likely overcorrects for the multiplicity issue – we view that added conservatism as a strength rather than a problem. Crucially, none of our central findings — specifically those related to the preregistered hypotheses concerning the effect of Perspective — are impacted by the Benjamini-Hochberg corrections or a separate Bonferroni correction for confirmatory tests only.

We have further clarified in our manuscript which effects were part of our preregistered confirmatory tests. Full ANOVA tables including the corrected p - and alpha-values (both uncorrected and adjusted) are reported in the online supplement.

We consider this analysis to be consistent, accessible, and robust (which does not mean other analysts might not do it differently – although they will inevitably observe the same results in the end). We reiterate that the litmus test of any effect is its replicability – given that we have replicated the crucial effect of perspective in 4 different studies, we are not terribly concerned about the multiplicity issue or which particular adjustment is preferable.

Pre-registration

The authors claim that their studies were pre-registered. While pre-registrations were made for all of the studies, the hypotheses presented in the paper do not match the ones that were actually pre-registered. What is described as the “Perspective hypothesis” (H2-H7) in the paper only occurs in pre-registration for the fourth study, which was presumably conducted after the first round of reviews. This is the only hypothesis in any of the pre-registrations that clearly sets forth a causal hypothesis in line with the narrative of the

paper. None of the three other pre-registrations contain any similar hypothesis – they are all correlational. The researchers have added an appendix describing which of the pre-registered hypotheses they claim test the hypotheses that are described in the paper, but this is in reality post hoc modification of hypotheses that undermines the claim that the hypotheses were pre-registered. While it can be acceptable to alter hypotheses or add new ones after pre-registration, this must be transparently and honestly acknowledged.

Response:

We respectfully disagree with the characterization of our pre-registrations and their relation to the hypotheses reported in the manuscript. All studies in this manuscript were pre-registered and the hypotheses described as H2-H7 reflect the same underlying theoretical prediction, despite the slight variation in phrasing across pre-registrations and experiments.

As we noted above in response to a similar comment, the interpretability of a hypothesis as causal is not determined by its linguistics *but exclusively by the experimental design and how the study was conducted*. There is no question that we examined causal effects of an experimental variable and we are unable to yield on this matter.

That said, we have now included all pre-registered hypotheses across all studies in the main body of the manuscript. We felt that the overview figure provided at the previous round made the paper more accessible, but we agree that the figure was not as precise and exhaustive as the table of all hypotheses that we include now.

Conceptual precision

Claims that are made in the paper should be accurate and precise. For example, the authors were asked what evidence there is that people's conceptions of truth and honesty have undergone significant changes in recent decades. The rebuttal letter contains a vague response ("The sentences in question serve to situate our work within the broader literature and establish the context for our study. They do not present new claims but rather summarize existing research to frame our contribution") but no answer to the actual question has been provided in the rebuttal letter or the paper. It is one thing to say that research has shown that this is a problem today and another to say that research has demonstrated changes over time.

Response:

We appreciate the reviewer's concern and upon further examination agree that our original framing could have made the point more explicitly. While the manuscript cited relevant literature throughout,

we acknowledge that we did not clearly state that this work demonstrates the changes over time in how truth and honesty are understood.

We have now revised the introduction to explicitly link the cited studies to this broader claim, clarifying that the literature documents a shift in these conceptions over recent decades. The following can be found on lines 102 through 106: “[...] Aroyehun et al. (2025) examined speeches of congress from 1879 until 2022 and discovered that the use of evidence-based language has steadily declined after reaching a peak in the mid 1970s: There has been a shift from evidence to intuition in congressional rhetoric. This shift has been associated with an increase in partisan polarisation in Congress and society at large.”

REVIEWERS' COMMENTS:

Reviewer #3 (Remarks to the Author):

In this timely and theoretically interesting manuscript, the researchers tested whether different perspectives on honesty – a focus on beliefs/authenticity or on facts – shaped people's responses to norm violations. The research tested and found support for the hypothesis that "when people are directed to rely on sincerity rather than accuracy as the main criterion of honesty, they became more tolerant of democratic norm violations."

This manuscript was characterized by a number of strengths. As noted above, the research is timely – it is of considerable interest and importance given shifting norms about societal discourse and the "backsliding" democratic norms. The work is also theoretically novel – it is grounded in literature on motivated reasoning and perspective taking, but moves that literature into a new direction. The research is methodologically sound, and the data analyses are rigorous, comprehensive, and well-reported. Taken together, the strengths of this research indicate it could make a meaningful contribution to existing literature.

Response: We thank the reviewer for their positive impressions of the strengths of our paper.

Nonetheless, this manuscript was also marked by some meaningful limitations. Perhaps the most important limitation was a conceptual and interpretive one. The introduction and justification for the study asserts that the problem explored in this study – an emphasis on "belief-speaking" yielding acceptance for norm violation – is one that exists primarily on the right. In fact, the introduction paints a picture that this pattern exists *only* on the right. Although it is not useful to engage in "both-sideism" merely for the sake of doing so, I was struck by the notion that the left also engages in "belief-speaking" and that the introduction does not seem to recognize that. One anecdotal example: the oft-maligned notion that people should "speak their truths" is prominent among the left but not the right. Consistent with the notion that belief-speaking might exist on the left and right, research also indicates that liberals and conservatives similarly reject scientific evidence when it does not fit their worldview (see Washburn and Skitka), engage in confirmation bias, and engage in motivated reasoning. The literature review does not meaningfully explore the possibility that belief-speaking exists (perhaps in different forms) across the political spectrum, focuses narrowly on research documenting the rigidity of the right, and ignores research documenting motivated reasoning as a human rather than a politicized tendency. Still, an open-minded reader might be willing to accept the notion that there is ideological asymmetry in belief-speaking if the data bore that out. Alas, when one gets to the results of this studies, the data shows no such thing. The researchers do a deep-dive on these tendencies among participants on the right, but do no such analysis for participants on the left. Taken together, the data show, as the manuscript confesses, "This effect is present regardless of a

person's political orientation or pre-existing epistemic preferences." Afterward, the discussion section of the manuscript returns to the notion that this is a problem of the right . Furthermore, given the centrality of political orientation in your introduction and discussion, it is notable that you only include the variable as a covariate in one of the four studies.

So, what could one do in response to this critique? In the introduction, you could either remove the focus on political orientation as a driver of this phenomenon, or you could broaden the focus to include the possibility that (as you find) this phenomenon exists along the entire political spectrum. If these changes feel inappropriate or misguided because they might lead "hypothesizing after the results are known", then the correction could exist within the discussion section. From my perspective, your fundamental finding IS interesting and important, even if you made the change to describe it as a tendency that exists among people in general and as one that is a risk to democratic norms.

Response: We thank you for this thorough feedback and for raising the issue of people on both sides of the political spectrum engaging in motivated reasoning. We agree that we gave short shrift to left-wing populism in the previous version (only one brief footnote). We have therefore extended the discussion of left-wing populism in the introduction and now justify it more clearly why our work focuses on right-wing political extremism.

The other limitation is a relatively minor one. There is a lack of clarity when it comes to the method. The manuscript reports four studies in one combined method and results section. Given the clear and meaningful similarities between the studies that seems like a reasonable choice. But the justification for each of the studies and their differences is difficult to ascertain - which makes it more difficult to make sense of the findings and would undermine attempts at replication. This could be cleared up with a paragraph and/or table that clarifies the distinctions between the studies and their purposes.

Response: We thank the reviewer for this helpful suggestion. We agree that while the studies were intentionally presented together due to their shared design and theoretical focus, the rationale for each study and the distinctions between them were not sufficiently explicit in the original manuscript. To address this, we have added clarifying content in the "Methods" section. We believe these revisions clarify the overall structure of the research and make the findings easier to interpret and replicate.

In sum, the core finding of the research was theoretically interesting, timely, and interesting. The research was well-designed (despite a lack of clarity), and the analysis was appropriate and comprehensive. The manuscript, nonetheless, would be strengthened if it chose a more consistent, more accurate, and more theoretically grounded approach to motivated reasoning across the political spectrum rather than presenting it as a problem of the right.

Response: We are grateful that the reviewer saw considerable merit in our work and we have prepared a revision that we hope successfully addresses the remaining problems.